# Statins Induce Locomotion and Muscular Phenotypes in *Drosophila melanogaster* That Are Reminiscent of Human Myopathy: Evidence for the Role of the Chloride Channel Inhibition in the Muscular Phenotypes

**DOI:** 10.3390/cells11223528

**Published:** 2022-11-08

**Authors:** Mohamed H. Al-Sabri, Neha Behare, Ahmed M. Alsehli, Samuel Berkins, Aadeya Arora, Eirini Antoniou, Eleni I. Moysiadou, Sowmya Anantha-Krishnan, Patricia D. Cosmen, Johanna Vikner, Thiago C. Moulin, Nourhene Ammar, Hadi Boukhatmi, Laura E. Clemensson, Mathias Rask-Andersen, Jessica Mwinyi, Michael J. Williams, Robert Fredriksson, Helgi B. Schiöth

**Affiliations:** 1Department of Surgical Sciences, Division of Functional Pharmacology and Neuroscience, Biomedical Center (BMC), Uppsala University, Husargatan 3, 751 24 Uppsala, Sweden; 2Department of Pharmaceutical Biosciences, Uppsala University, 751 24 Uppsala, Sweden; 3Faculty of Medicine, King Abdulaziz University and Hospital, Al Ehtifalat St., Jeddah 21589, Saudi Arabia; 4Faculty of Medicine, Department of Experimental Medical Science, Lund University, Sölvegatan 19, BMC F10, 221 84 Lund, Sweden; 5Institut de Génétique et Développement de Rennes (IGDR), Université de Rennes, CNRS, UMR6290, 35065 Rennes, France; 6Department of Immunology, Genetics and Pathology, Uppsala University, 752 37 Uppsala, Sweden

**Keywords:** statins, fluvastatin, statin-induced myopathy, Hmgcr, skeletal muscle chloride channel, ClC-a, CLC-1, myopathy, locomotion, sarcomere, mitochondrial dysfunction, lipotoxicity, Pkcdelta (Pkcδ), PKCtheta (PKCθ), chelerythrine, *Drosophila melanogaster*

## Abstract

The underlying mechanisms for statin-induced myopathy (SIM) are still equivocal. In this study, we employ *Drosophila melanogaster* to dissect possible underlying mechanisms for SIM. We observe that chronic fluvastatin treatment causes reduced general locomotion activity and climbing ability. In addition, transmission microscopy of dissected skeletal muscles of fluvastatin-treated flies reveals strong myofibrillar damage, including increased sarcomere lengths and Z-line streaming, which are reminiscent of myopathy, along with fragmented mitochondria of larger sizes, most of which are round-like shapes. Furthermore, chronic fluvastatin treatment is associated with impaired lipid metabolism and insulin signalling. Mechanistically, knockdown of the statin-target *Hmgcr* in the skeletal muscles recapitulates fluvastatin-induced mitochondrial phenotypes and lowered general locomotion activity; however, it was not sufficient to alter sarcomere length or elicit myofibrillar damage compared to controls or fluvastatin treatment. Moreover, we found that fluvastatin treatment was associated with reduced expression of the skeletal muscle chloride channel, ClC-a (*Drosophila* homolog of *CLCN1*), while selective knockdown of skeletal muscle ClC-a also recapitulated fluvastatin-induced myofibril damage and increased sarcomere lengths. Surprisingly, exercising fluvastatin-treated flies restored ClC-a expression and normalized sarcomere lengths, suggesting that fluvastatin-induced myofibrillar phenotypes could be linked to lowered ClC-a expression. Taken together, these results may indicate the potential role of ClC-a inhibition in statin-associated muscular phenotypes. This study underlines the importance of *Drosophila melanogaster* as a powerful model system for elucidating the locomotion and muscular phenotypes, promoting a better understanding of the molecular mechanisms underlying SIM.

## 1. Introduction

Cardiovascular diseases (CVDs) are the leading cause of morbidity and mortality globally, accounting for approximately 17.9 million patients dying each year, with hypercholesteremia as the main risk factor [1,2]. Statins, 3-hydroxy-3-methylglutaryl-CoA reductase (HMGCR) inhibitors, are widely prescribed to lower cholesterol and low-density lipoprotein (LDL) and thus reduce CVDs mortality and morbidity [3,4,5,6,7]. Although relatively safe, multiple reports have shown that due to statin intolerance, more than 30% of patients discontinue statin medications, leading to increased risks for CVD events [8,9,10,11]. Statin intolerance is mainly due to muscle problems, including myalgia (without creatine kinase (CK) elevation), myositis (CK 10-fold within the upper limit of normal, ULN), and rhabdomyolysis (CK elevation over 10-fold the ULN along with reddish or brown urine) [12], which are grouped under the term statin-induced myopathy (SIM) [13,14,15]. In the United States, a recent internet-based survey has indicated that up to 60% of former statin-treated patients have reported an incidence of SIM-related sequelae, with muscle problems as the primary reason for statin discontinuation [10]. More importantly, SIM-related muscle damage has been reported long after the cessation of statins [9,16]. Indeed, statin alternatives do not offer better prevention and treatment for CVDs compared to statins [17]. Therefore, identifying the root cause of SIM is very crucial, and a recent simulation study has shown that improving statin adherence by 50% would lower CVD-related deaths [18,19].

Despite extensive research, the aetiology of SIM remains enigmatic. One theory has attributed SIM to a reduction in muscular membrane sarcolemmal cholesterol levels [20]. Yet, blocking cholesterol synthesis by squalene synthase inhibitors does not cause myopathy [21]. Another hypothesis has linked SIM to isoprenoid depletion, one of the components of the mevalonate pathway which is diminished by the inhibition of HMGCR. Depleted isoprenoid levels result in higher intracellular calcium concentrations, which can trigger cell death in muscles [22,23,24]. However, a study by Laaksonen et al. has reported that statins do not reduce isoprenoid synthesis in the muscles significantly [25]. Other studies have demonstrated that, due to HMGCR inhibition, statins lead to coenzyme Q10 (CoQ10) depletion and mitochondrial dysfunction, with subsequent muscle problems [26]. Nevertheless, shreds of evidence have indicated that muscular CoQ10 levels are not changed in most statin-treated patients, which could explain why CoQ10 supplementation does not mostly alleviate SIM [25,26,27,28,29,30,31,32,33,34]. Moreover, a recent study indicates that statins, except for fluvastatin, induce mitochondrial dysfunctions by binding to mitochondrial Complex III [35]. However, other scholars have pointed out that statins do not cause changes in mitochondrial functions or morphology in muscles of statin-treated patients [36,37,38] and clinical research has indicated that simvastatin can improve mitochondrial functions in cardiomyocytes [39], which all have questioned the primary role of mitochondrial dysfunction in SIM.

Apart from the mevalonate pathway, a reduction in insulin secretion and sensitivity by statins has been also proposed as an underlying factor for SIM [40,41,42]. It has been also reported that statin treatment is associated with the inhibition of skeletal muscle chloride channel 1, CLC-1 [43,44,45,46], which is the key feature of myotonia [47,48]. Nevertheless, whether statin-associated inhibition of CLC-1 can cause myopathy phenotypes and the mechanistic pathway underlying statin-associated inhibition of CLC-1 are still poorly understood.

*Drosophila melanogaster* (fruit fly) provides an ideal system to understand muscle disorders, including myopathy with potential application for human conditions and many features of morphology and physiology of skeletal musculature in fruit flies are remarkably similar to humans [49,50,51,52,53]. We hypothesize that statin might induce myopathy phenotypes through direct inhibition of HMGCR or indirect inhibition of skeletal muscle chloride channel or both. Since statin-target HMGCR (in flies, HMGCR is denoted as Hmgcr) is highly conserved in both humans and flies [54,55,56], we used fruit flies to employ genetic and pharmacological approaches to investigate whether Hmgcr inhibition in skeletal muscles gives rise to myopathy-like phenotypes. Moreover, we examined whether statin inhibits skeletal muscle chloride channels and whether muscular inhibition of this channel contributes to myopathy phenotypes.

## 2. Materials and Methods

### 2.1. Fly Stocks and Maintenance

Fly stocks mentioned in Table 1 were purchased from the Bloomington Stock Center (Bloomington, IN, USA) and used for the experiments. W1118 wild-type flies were used from our in-house stock. The wild-type CSORC flies were obtained by crossing CantonS and OregonR-C flies, both types of flies were ordered from Bloomington Stock centre (Bloomington, IN, USA). All flies, unless otherwise stated, were maintained on enriched Jazz mix standard fly food (Fisher Scientific, Waltham, MA, USA). Flies were maintained at 25 °C in an incubator at 60% humidity on a 12:12 light:dark cycle. Virgin CSORC males aged 1–2 days were used in this study unless otherwise stated. In order to feed flies with drugs, a proper volume of the drug was mixed with warm food in a small plastic vial. Then, we allowed it to cool before we nurtured flies in these plastic vials containing food with/without the drug(s) for five days before experiments.

### 2.2. Setting Up the Crosses

To knockdown *Hmgcr* in muscles, virgin female *Mhc-Gal4* was crossed with male *UAS-Hmgcr RNAi* line, referred to as *Mhc-Gal4>UAS-Hmgcr RNAi*, while virgin female *w1118* crossed with male *UAS-Hmgcr RNAi*, referred to as *w1118>UAS-Hmgcr RNAi*, and virgin female *Mhc-GAL4* line crossed with male *w1118*, referred to as *Mhc-Gal4>w1118,* both are used as control groups. Only straight-wing flies were collected and used in the experiments. Most *Hmgcr^RNAi^* knockdown flies (*Mhc-Gal4>Hmgcr RNAi*) had a shorter lifespan, less than two days (Appendix A). We tried to overcome this short-life span problem by raising the flies at 25 °C, instead of 27 °C and collecting them immediately for the experiments, but the problem remained. Consequently, we used one-two-day-old flies for the experiments. To knockdown ClC-a in muscles, virgin female *Mhc-Gal4* was crossed with male *UAS-ClC-a RNAi* line, referred to as *Mhc-Gal4>UAS-ClC-a RNAi*, while virgin female *w1118* crossed with male *UAS-ClC-a RNAi*, referred to as *w1118>UAS-ClC-a RNAi*, and virgin female *Mhc-GAL4* line crossed with male *w1118,* referred to as *Mhc-Gal4>w1118*, both are used as control groups. Only straight-wing flies were collected and used in the experiments. To knockdown *Pkcdelta* in the skeletal muscles, virgin female *EDTP-Gal4* was crossed with male *UAS-Pkcdelta RNAi*, referred to as *EDTP-Gal4>UAS-Pkcdelta RNAi*, while virgin female *w1118* crossed with male *UAS-Pkcdelta RNAi*, referred to as *w1118>UAS-Pkcdelta RNAi*, and virgin female *EDTP-GAL4* line crossed with male *w1118,* referred to as *EDTP-Gal4>w1118*, both are used as control groups. Unless otherwise stated, crossed flies were raised at 25 °C until the adults emerged; once collected, adults were raised at 27–29 °C for appropriate times. In all assays, F1 progeny was used.

### 2.3. Exposure Concentration

For the maximum concentration exposure assessment, we first looked for the fluvastatin concentrations used in the literature. The fluvastatin concentrations used before were 0.05, 0.5 and 1 mM [55] and 5 mg/µL = 11.5 mM [57] without reported toxic effects. Since 0.5 and 1 mM concentrations have been reported to significantly decrease mean activity and enhance sleep duration in *Drosophila* [55], we assessed whether these concentrations are within the reported toxic fluvastatin dose in mice and in relation to humans’ mean plasma concentration of 40 mg fluvastatin, the daily recommended dose for adults. According to the literature, the fly adult male weight of 0.88 mg [58] and daily intake of 1.5 µL [59,60]. Next, we selected a range of concentrations of 0.05–40 mM and then calculated how much fluvastatin mg/kg/day in these concentrations corresponded to. After that, we compared them to fluvastatin doses in mg/kg/day used in the toxicity studies in mice (https://www.accessdata.fda.gov/drugsatfda_docs/label/2022/201635s029lbl.pdf, accessed on 1 August 2022). According to our calculations, we found that fluvastatin concentrations of 20 mM (14.7 mg/kg/day) and 40 mM (29.5 mg/kg/day) are equivalent to reported toxic cancerogenic doses in mice (15 and 30 mg/kg/day), which are 2 and 7 times the mean human plasma drug concentration after a 40 mg oral dose. Moreover, the regular fluvastatin dose in humans is 40 mg/day and the maximum dose is 80 mg/day; thus, considering the average human male weight is 70 kg, the daily doses per kg are 0.57 and 1.1 mg/kg/day, respectively. Therefore, we concluded that 0.5 mM (0.3 mg/kg/day) and 1 mM (0.74 mg/kg/day) are within the therapeutic dose range and suitable for our study.

### 2.4. Quantification of Climbing Ability Using Forced-Climbing System

To examine the effect of fluvastatin on the climbing ability of the flies, we used the Forced-Climbing System as described by Moulin et al. [61]. In short, this system comprises two MultiBeam Activity Monitor, MB, (TriKinetics Inc., Waltham, MA, USA) placed vertically in Vortexer Mounting Plate, VMP, (TriKinetics Inc., Waltham, MA, USA) see Figure 1.

Flies were anaesthetised using CO_2_ and placed in individual glass tubes covered with cotton on one end and the other end with a black plastic lid. The glass tubes were loaded vertically into the MB5 with the side containing cotton on the bottom. The flies were agitated at a regular interval (4 s per minute) by the VMP for six hours, starting at 13:00 and the light was kept turned on for the entire experiment. After each agitation, flies fell to the bottom of the tubes and then, due to their natural negative geotaxis, flies climb up the tube against gravity and crossed the 17 IR beams.

We have optimized the Forced-Climbing System to monitor not only fly moves but also speed and positions in the tubes (representing how far can flies climb up). Specifically, we developed a MATLAB routine that allows us to record the flies’ average moves per hour, speed per hour and positions. In short, first, we retrieved the raw files from MB5 software after each run. Then, we applied the following equations:

Hourly locomotor movement (moves) is calculated by averaging the hourly beam-to-beam transition recorded generated from the flies’ activity in the course of the experimental time
$$\mathrm{Moves}\;\mathrm{per}\;\mathrm{hour}=\frac{\sum Beam\;to\;beam\;transition\;for\;the\;n^{th}\;hour\;in\;EH\;}{3600-\left(x*60\right)}$$

The hourly maximum position reached hourly (positions) is determined by averaging the maximum position reached by the flies in each hour
$$\mathrm{Position}=\frac{\sum Position\;of\;fly\;at\;any\;particular\;instant\;for\;the\;n^{th}\;hour\;in\;EH\;}{3600-\left(x*60\right)}$$

Hourly locomotory speed (Speeds) is the summation of the difference between the position of fly at *y^th^* second and position of fly at *x^th^* second divided by *n* for the entire course of experimental time
$$\mathrm{Speed}\;\mathrm{per}\;\mathrm{hour}\;=\left(\left[\sum \frac{Fly\;position\;at\;y^{th}\;sec-x^{th}\;sec}{y-x}\right]for\;each\;hour\right)*3\;mm/sec\;$$
where

*EH*—Experimental hours

*TSEA*—$\left(EH*3600\right)-\left(EH*x*60\right)$

*x*—Number of seconds VPM agitated

*y*—User-determined seconds (>*x* and <60) and by default 10

The length between each IR is 3 mm.

I.Acute and chronic fluvastatin treatment

Fluvastatin sodium hydrate (MW: 433.45, 10 mg, Sigma, Darmstadt, Germany, #SML0038) was dissolved in water to make the stock solution. Generally, unless otherwise stated, we calculated the required volume of the fluvastatin and then mixed it with an appropriate volume of food to make the final concentration.

To determine the chronic and acute effects of fluvastatin on the flies, one–two-day-old virgin male CSORC flies were collected from the stock. For the control group, flies were fed normal food and then on the fifth day, we starved them for one hour before transferring them onto vials in which 3 mm Whatman filter paper immersed with 100 µL 5% (wt/vol) sucrose alone was placed in the bottom for three hours; for the fluvastatin group (five days), we fed flies with 0.5 mM fluvastatin and then on the fifth day, we fed them with 5% (wt/vol) sucrose alone; and for the fluvastatin group (three hours), flies were fed with normal food for five days and then on the fifth day, we fed them with 0.5 mM fluvastatin for three hours (Table 2).

II.Chronic fluvastatin 0.5 and 1 mM treatment

One–two-day-old virgin male CSCOR flies were collected for the experiments. In the control, 0.5 and 1 mM fluvastatin groups, fluvastatin powder was solubilized in water and mixed into the regular fly food, at concentrations of 0 (control), 0.5, or 1 mM. Then, flies were fed with food for five days before the experiments.

III.Chelerythrine and fluvastatin treatment

Chelerythrine chloride (CC) was purchased from Sigma, Germany #C2932. Virgin male CSORC flies of one-two day old were used. In the control and the control with CC groups, flies were fed normal food for five days, while in the fluvastatin group and the fluvastatin with CC group, flies were fed food mixed with 0.5 mM fluvastatin for five days. On the fifth day, we starved them for one hour before transferring them onto vials in which 3 mm Whatman filter paper was placed in the bottom. Then, into the filter paper, we applied 100 μL (sucrose alone) for the control group; 100 μL of 100 μM CC for the control with CC group; 100 μL of 0.5 mM for the fluvastatin group; and 100 μL of 100 μM CC for the fluvastatin with CC group. We used 5% (wt/vol) sucrose solution as a vehicle for the CC and fluvastatin feeding on the fifth day (Table 3).

### 2.5. Quantification of General Locomotion Activities Using Drosophila Activity Monitoring System (DAMS)

To examine the fluvastatin effect on the general locomotion activity of the flies, we used DAMS (Drosophila Activity Monitoring System) (TriKinetics Inc., Waltham, MA, USA). All crossed flies were fed with normal food unless otherwise stated.

In 0.5 and 1 mM fluvastatin groups, we prepared a stock solution of 5 mM fluvastatin using water as a vehicle. Then, the calculated volume of fluvastatin (1 and 2 mL, respectively) was taken and mixed with food to have a final volume of 10 mL, while in the control group, food was mixed with 1 mL of water. Each Petri dish mixture was kept to cool down and solidify for one hour. Thereafter, for each group, 30 glass 5 mm × 65 mm tube tips were dipped into the corresponding mixture twice and then plugged with plastic black caps. One–two-day-old virgin male CSCOR flies were collected for the experiments. Then, each fly was transferred into each tube using a CO_2_ anaesthesia pad and a paintbrush. Finally, the other ends of the tubes were subsequently plugged with cotton, and the tubes were placed in the DAMS monitor horizontally.

For the preparation of the fluvastatin solution and CC solution, we used water as a vehicle. To check the effect of the fluvastatin on flies, we first fed CSORC flies with 1 mM and 0.5 mM fluvastatin and then we run DAMS. To look out whether CC can rescue the fluvastatin phenotype, we divided CSORC flies into four groups: for the fluvastatin group, we prepared 0.5 mM concentration from the stock concentration mixed with 10 mL hot food in a 12 mL Petri dish so that the total volume of 12 mL is obtained; for the fluvastatin with CC, we prepared 0.5 mM both fluvastatin and cc mixed with 10 mL hot food in a 12 mL Petri dish so that the total volume of 12 mL is obtained. For the CC group, we added only CC and for the control, we added only the food (Table 4). Each Petri dish mixture was kept to cool down and solidify for one hour. Thereafter, for each group, 30 glass 5 mm × 65 mm tube tips were dipped in the corresponding mixture twice and then plugged with plastic black caps.

For the chloride channel blockers, 2-(4-chlorophenoxy) propionic acid (CPP) (50 mg, R750298, Sigma, Taufkirchen, Germany) and 9-anthracenecarboxylic acid (A9C) (50 mg, A89405, Sigma, Germany) were obtained from Sigma-Aldrich^TM^. We used 70% ethanol as a vehicle for CPP and A9C. We divided the flies into three groups (control, 1 mM CPP, 1 mM A9C), each group consists of 32 flies (Table 5). CSPRC virgin males were collected and transferred into the tube in the same way mentioned above. We recorded their general activity, mean activity and activity counts/30 min all for five days. For the crossed flies with GAL4, we used normal food and flies aged was 1–5 days before experimentation, unless otherwise mentioned.

Finally, all monitors were placed in the incubator for five days at 25 °C, and 60% humidity on a 12:12 light:dark cycle. On the sixth day, the data for the flies’ general activity for five days were collected from the DAMSystem software and analysed using MATLAB, and GraphPad Prism.

### 2.6. Transmission Electron Microscopy

i.Morphology

Flies were first cool-anaesthetized by chilling in crushed ice until they stop moving to ensure relaxation of the muscles before dissection. Five flies per group taken from vials were cool-anaesthetized by chilling in crushed ice until they stopped moving. Next, the heads of flies were first cut with a sharp razor and then, cross-sections from femur segments of legs were taken and fixed for morphology. The EM preparation was performed by BioVis Core Facility/Electron microscopy at IGP, Uppsala University, Uppsala, Sweden. In short, the legs were dissected and the dissected tissues were Fixed in 2.5% glutaraldehyde and 1% paraformaldehyde in 0.1 M sodium cacodylate buffer (pH 7.6) overnight at 4 °C. Then, we rinsed it in 0.1 M sodium cacodylate buffer (pH 7.6) for 10 min, Post-fix in 1% osmium tetraoxide in 0.1 M sodium cacodylate buffer (pH 7.6) for 1 h, dehydrated in increasing concentrations of EtOH (50%, 70%, 95%, 100%) for 15 min, embedded in LR white and 100% EtOH (1:1) for 60 min, changed to 100% LR white overnight at 4 °C, transferred to gelatine capsules with 100% LR white for 2–3 h, polymerized at 55 °C for 48 h, sectioned ultrathin sections (~60 nm) on an ultramicrotome (Leica EM UC7), placed sections on a cu slot grid (Ted Pella, Redding, CA, USA) coated with Formvar, contrasted with 2% Uranyl acetate for 5 min, washed extensively with water, again contrasted with 3% lead citrate (Reynold’s, Merck, Darmstadt, Germany) for 2 min, again washed extensively with water, air dried for 15 min and finally imaged in Technai™ G2 Spirit (ThermoFisher/FEI, Eindhoven, The Netherlands) electron microscope at 80 kV using a Lab6 filament and a Gatan Orius SC200 camera.

ii.Immunogold labelling

In the fluvastatin group, flies were fed with 1.0 mM fluvastatin in the same way mentioned above, while in the control group, flies were fed with normal food, both groups for five days. Five flies per group taken from vials were cool-anaesthetized by chilling in crushed ice until they stopped moving to ensure the relaxation of the muscles before dissection. Next, the heads of flies were first cut with a sharp razor and then, cross-sections from femur and tibia segments of legs were taken and fixed for immunogold labelling. The EM preparation was performed by BioVis Core Facility/Electron microscopy at IGP, Uppsala University. In short: the femur of the legs were dissected and dissected tissue was fixed in 4% paraformaldehyde and 0.05% glutaraldehyde in 0.1 M sodium cacodylate buffer (pH 7.6) O.N. at 4 °C, rinsed in 0.1 M sodium cacodylate buffer (pH 7.6) for 10 min, dehydrated in increasing concentrations of EtOH (50%, 70%, 95%, 100%) for 15 min, embedded in LR white and 100% EtOH (1:1) for 60 min, changed to 100% LR white O.N. at 4 °C, transferred to gelatine capsules with 100% LR white for 2–3 h and polymerized at 55 °C for 48 h. Legs were then sectioned in ultrathin sections (~60 nm) on an ultramicrotome (Leica EM UC7), and placed on a Ni 200-mesh grid (Ted Pella) coated with formvar. Free aldehyde groups were blocked with 50 mM glycine in PBS for 15 min. Grids were transferred to blocking solution for goat secondary reagents (Aurion, Wageningen, The Netherlands) for 30 min, washed in 0.1% BSA-C (Aurion, Wageningen, The Netherlands) 3 × 5 min. Primary antibody Rabbit Anti-ClC-2 Antibody (Abcam, Amsterdam, The Netherlands, #ab192506) (ClC2, 1:500) was then added and sections were incubated O.N. at 4 °C and then washed in 0.1% BSA-C (Aurion, Wageningen, The Netherlands) 6 × 5 min. A secondary antibody (goat anti-rabbit, 12 nm gold colloidal gold particle, Jackson Immuno Research Europe Ltd., Ely, UK) was then added for 60 min. Sections were then washed in 0.1% BSA-C (Aurion, Wageningen, The Netherlands) 6 × 5 min and washed in MQ Water 6 × 5 min. Sections were contrasted with 2% Uranyl acetate for 5 min and again washed extensively with water and air-dried. Pictures of the grids were then taken, and ImageJ was used to quantify immunogold particles.

iii.Chloride channel particles counting

The grid for each group obtained from immunogold labelling TEM was divided into an area of 33 × 10^6^ pixels (8.7 × 10^9^ micron). Next, we quantified the ClC-a gold particle in each area using ImageJ software (version 1.53Ki, National Institutes of Health, Bethesda, MD, USA) and calculated the statistical difference using GraphPad Prism. The type of the test is indicated in the caption of each figure.

iv.Measuring the sarcomere length

TEM micrographs with a scale bar of 2 or 5 µm have been selected randomly from the groups. The length between Z-lines of the sarcomere was measured in µm using ImageJ (version 1.53Ki, National Institutes of Health, Bethesda, MD, USA) from the top, middle and bottom of the sarcomere. Then, the significant difference was calculated using GraphPad Prism. The type of the test is indicated in the caption of each figure.

v.Measuring the mitochondria area

Measuring mitochondrial areas in TEM micrographs with a scale bar of 1, 2 and 5 µm was carried out using Image J (version 1.53Ki, National Institutes of Health, Bethesda, MD, USA) by manually tracing mitochondria from selected TEM micrograph. Surface area (mitochondrial size) is reported in squared micrometres. The type of the test is indicated in the caption of each figure.

vi.Counting the number of round-like shaped mitochondria

Counting the number of round-like shaped mitochondria in TEM micrographs with a scale bar of 1, 2 and 5 µm was carried out using Image J (version 1.53K, National Institutes of Health, Bethesda, MD, USA). The type of the test is indicated in the caption of each figure.

### 2.7. Quantification of Genes and Protein Expressions

i.RNA purification and cDNA synthesis

For the crossed flies, male flies were collected and allowed to age for five days at 27–29 °C per group and then stored at −80 °C before extraction. For CSORC flies, for the fluvastatin group, 100 virgin males were transferred onto a vial containing 5 mL of food mixed with a subsequent volume of fluvastatin stock solution sufficient to make 1 mM fluvastatin, and for the control group, another 100 flies were transferred onto a vial containing 5 mL of food mixed with an equivalent volume of water. Having allowed the flies to age for five days at 25 °C in the mixture, we extracted the RNA using the whole body, and then stored them at −80 °C before extraction. We used chloroform (Sigma #67663), isopropanol (Sigma #I9516) and triazole (Thermofisher #15596026), and the protocol was applied according to Williams et al. [62]. The level and purity of the RNA and cDNA were measured by nanodrop in a Multiscan machine (Thermo Scientific™) using μDrop™ Plate. The accepted Absorbance 260/280 for both, RNA and cDNA ranged from 1.9 to 2.1. The cDNA was synthesized by Thermal cyclers using 20 μL total volume: 9 μL of the extracted RNA, 10 μL RT buffer (Thermofisher, High-Capacity RNA-to-cDNA™ Kit #4387406) and 1 μL Reverse Transcriptase Enzyme (Thermofisher, High-Capacity RNA-to-cDNA™ Kit #4387406).

ii.Real-time qPCR

cDNA was diluted in a ratio of 1:30 using Milli-Q^®^ water and the level of genes of interest was measured using quantitative real-time polymerase chain reactions, qPCR. Primers were selected through FlyPrimerBank and ordered from ThermoFisher™ (Table 6).

Relative expression levels of the housekeeping gene (*RpL32*) and the genes of interest were measured using quantitative RT-PCR (qPCR), Bio-Rad iCycler Thermal Cycler Real-Time PCR Detection. Each plate well contained a total volume of 20 μL of the following: 11.52 μL Milli-Q^®^ water, 3.60 μL Taq buffer, 0.20 μL dNTPs, 0.1 μL primers, 1 μL DMSO (1:20), 0.5 μL SYBR Green (1:50,000) and 3 μL of cDNA of the gene of interest as described in Williams et al. [62]. All qPCR experiments were carried out in triplicates. The qPCR data were analysed using MyIQ 1.0 software (Bio-Rad, Hercules, CA, USA) and values of fold changes (ratio of delta CT value) for the samples were normalized with *RpL32′s* values. Outliers of CT ratio were checked out by Grubbs’ test and the significant outliers (*p* > 0.05) were excluded. GraphPad Prism was used to calculate the significance. The type of the test is indicated in each figure’s caption.

iii.Western Blotting

For fluvastatin groups, 10 virgin-CSORC males of one–two-day-old per each group were transferred onto a vial containing 5 mL of food mixed with a subsequent volume of fluvastatin stock solution sufficient to make 1 mM or 0.5 mM fluvastatin, and for the control group, another 10 virgin-CSORC male flies aged 1–2 days old were transferred onto a vial containing 5 mL of food mixed with an equivalent volume of water. We grew the flies in the mixture for five days and then stored them at −80 °C. After that, flies were homogenized in RIPA buffer (Thermofisher #89900) mixed with protease and phosphatase inhibitors (Thermofisher #78441), and then we loaded an equal amount of lysate (equivalent to 30 µg) into SDS-PAGE gel and blotted to the PVDF membrane. For measuring the protein expression of CLC-a, we ran a pair-wise sequence comparison to select the antibodies and the following antibodies were applied: for anti-CLC-a antibodies, Rabbit Anti-ClC-2 Antibody (Abcam, The Netherlands, #ab192506) at 1:500 dilution; β-Actin polyclonal Antibody (ThermoFisher #PA5-16914) at 1:1000 dilution. By using the mild stripping protocol from https://www.abcam.com/protocols/western-blot-membrane-stripping-for-restaining-protocol, accessed on 1 August 2022, the blots were stripped and re-probed in the same PVDF membrane. We used the secondary antibody Rabbit IgG (H + L) Cross-Adsorbed Secondary Antibody (ThermoFisher, HRP #A16104) at 1:1000 dilution. Enhanced reagent ECL kits from Biorad were used for detecting the signals. Signal detection was performed using ChemiDoc MP Imaging System, Bio-Rad. Relative intensity was calculated by normalizing the absolute intensity of the experimental group to the absolute intensity of β-actin using Image Lab and GraphPad Prism software. The *p*-value was calculated using GraphPad Prism. The type of the test is indicated in each figure’s caption.

## 3. Results

### 3.1. Chronic Fluvastatin Treatment Induces Lowered General Locomotion Activities and Climbing Ability

Statin treatment has been reported to induce myopathy in humans [9,45], and myopathy-like phenotypes in mice [63], rats [64], goats [65] and zebrafish and chicken [66,67,68]. Thus, we first asked whether statins induce phenotypes reminiscent of myopathy in the fruit fly. Previous studies showed that fluvastatin is effective in flies and able to inhibit Hmgcr activity in the mevalonate pathway [69,70]. Additionally, we previously demonstrated that fluvastatin treatment for five days reduces the mean activities and promoted sleep duration [55]. Therefore, we examined the effect of fluvastatin on the climbing ability and general locomotion activity using wild-type CSORC flies as they are commonly used in locomotion and climbing assays [61,71]. We employed the *Drosophila* Activity Monitoring System (DAMS) for monitoring general locomotion activities and improved the Forced-Climbing System described by Moulin et al. [61] to record different aspects of fly climbing ability including the hourly average of moves, speed and positions (see Materials and Methods).

First, we tested the chronic effect (5 days) of 0.5 and 1 mM fluvastatin concentrations on the climbing ability and general locomotion activities. In Forced-Climbing activity, compared to the controls, both, 1.0 mM and 0.5 fluvastatin-treated flies showed significantly lowered climbing ability including the average moves and speed. However, the average position was significantly reduced only in 1 mM fluvastatin-treated flies (Figure 2A–C). If we compared both concentrations, 1 mM fluvastatin had a more significant decline in the average speed and position than 0.5 mM (Figure 2B,C). In DAMS, both 1.0 mM and 0.5 fluvastatin-treated flies displayed significantly lowered general locomotion activities (activity count/30 min, and total activity) while the activity while awake was not altered (Figure 2D and Appendix A). However, acute fluvastatin treatment of CSORC flies for three hours did not change the climbing activity, particularly the average moves (Appendix A). In total, these results indicate that chronic fluvastatin treatment impairs climbing and locomotion activities. In line with these results, statin treatment has been reported to markedly reduce zebrafish movement [66] and reduce the running capacity in mice [72].

### 3.2. Chronic Fluvastatin Treatment Induces Phenotypic Changes in Myofibrils and Mitochondria and Is Associated with Reduced Muscle Regeneration

To determine whether the lowered locomotion and climbing activities associated with fluvastatin treatment were a result of muscle problems, we treated CSORC flies with 1 mM fluvastatin for five days and examined the morphology of the skeletal muscles of the femur segments of legs using transmission electron microscopy (TEM). Interestingly, in most TEM micrographs, compared to the control group, the skeletal muscle ultrastructure of the fluvastatin-treated flies exhibited more remarkable skewing, displacement, or contortion of myofibrils, specifically, misorientation of sarcomeres, Z-line streaming or/and loss of Z-line and distended T-tubules together with a few vacuoles and a rare occasion of myelin figures, which are circular scroll-like myofilaments, that are similar to toxic myopathy in humans [73] (Figure 3A–C).

The sarcomere is the functional contractile unit of myofibril delimited by two Z lines. Thus, sarcomere size correlates with the diameter of myofibrils and muscle fibres and their contraction strength as well [74,75,76,77,78,79]. In addition, alteration in sarcomere integrity or/and size affects the whole muscles and therefore sarcomere length/myofibril diameter has been taken as a hallmark of myopathy-related diseases [75,76,77,78,79,80] and in SIM [65,66,68,81,82,83,84]. Accordingly, alterations in myofibril ultrastructure and sarcomere length were taken as reminiscent of myopathy in this study. In this context, strikingly, the fluvastatin-treated flies displayed a larger sarcomere length comparable to the control group (Figure 3D). However, at this point, we did not know exactly whether the increased sarcomere length is caused by longer actin and myosin filaments or due to the greater spacing in the I-Band or A band regions. Therefore, it would be interesting to investigate this using immunohistochemistry and confocal microscopy. Paradoxically, when fluvastatin-treated flies were exercised in the Forced-Climbing System for six hours, the sarcomere length was comparable to the control, although the ultrastructure alteration of myofibril was still obvious (Appendix A). In agreement with these results, treating rats with a low dose of fluvastatin led to histological alteration at the myofibril and larger fibre diameters compared to controls [84]. In goats, skeletal muscles treated with lovastatin resulted in longer fibre diameter [65]. In humans, reports have pointed out that statins cause damage to muscle fibres, wider sarcomeres, loss of Z-lines, separation of myofibrils and distended T-tubules [81,82]. However, in Zebrafish, statins have been reported to induce shortened sarcomeres without affecting muscle fibre integrity [68].

At mitochondrial levels, in most TEM micrographs, the dissected leg muscles of fluvastatin-fed CSORC flies also exhibited aberrant mitochondrial morphology such as scattered abnormal mitochondria of various shapes, including round-like shapes, and of different sizes, including a few occasions of giant mitochondria (Figure 3A,B). In contrary, in most TEM micrographs, the control group displayed arranged, aggregated and elongated-shaped mitochondria that occasionally isolate neighbouring myofibrils (Figure 3A,B). Given the importance of size and the round shape of the mitochondria in muscle pathologies [85,86,87,88], we attempted to quantify surface area (size) and count the number of the round-like shaped mitochondria using ImageJ. Interestingly, in contrast to the control group, the fluvastatin-treated flies showed significantly larger mitochondria sizes (Figure 3E) and the number of round-like shaped mitochondria was remarkably higher (Figure 3F). In line with these results, several publications have pointed out that statins cause enlarged and round-like shaped mitochondria [89,90,91,92,93].

In order to examine the effect of fluvastatin on muscle development and regeneration, we measured the expression of *Mhc* (Myosin heavy chain and *Drosophila* homolog of MHC), *Mef2* (myocyte enhancer factor-2 and Drosophila homolog of *MEF2*) and *zfh1* (Zn finger homeodomain 1 and *Drosophila* homologue of *ZEB2*) due to their importance in muscle development, differentiation, and regeneration and since their disruption gives rise to muscle problems in flies, including myopathy [80,94,95,96,97,98,99,100,101]. We fed CSORC flies with 1mM fluvastatin mixed with food for five days while in the control group, flies were fed with food only. Then, we measured the expression of the genes using qPCR. Notably, CSORC flies treated chronically with 1.0mM fluvastatin showed a significantly reduced expression of *Mhc*, *p* = 0.0005, *Mef2*, *p* = 0.0019, and *zfh1*, *p* = 0.0042, suggesting that fluvastatin may impair muscle development differentiation in flies (Appendix A). Consistent with these effects, recent reports have demonstrated that statins have detrimental effects on the proliferation, differentiation, and regeneration of human skeletal muscle cells [102] and are associated with a considerable reduction in myosin heavy chain 1 protein and mRNA levels [49] and a significant decline in *MEF2* expression in muscle biopsies of statin-treated patients [45]. *zfh1* is essential for skeletal muscle differentiation [103]; thus, it would be interesting to examine the effect of statins on *ZEB2* on muscle biopsy from patients with SIM.

### 3.3. Chronic Fluvastatin Treatment Is Associated with Impaired Lipid Metabolism and Insulin Signalling

Mitochondrial damage and lipotoxicity are interrelated [104]. Several publications have shown that impaired lipid metabolism contributes to SIM [102,104]. Accordingly, we performed a transcriptome screen for the important genes involved in lipid metabolisms. Interestingly, most genes encoding enzymes that are involved in lipid metabolism were significantly downregulated in fluvastatin-treated flies (Appendix A and Table 7). Of importance, *Lpin* (*Drosophila* homolog of *LPIN1*) was highly downregulated in the fluvastatin group (*p* < 0.0001) (Appendix A and Table 7). *Lpin* encodes a phosphatidate phosphatase, which is responsible for the conversion of phosphatidic acid to diacylglycerol (DAG) during triacylglycerol (TG) biosynthesis. In mice, loss of its homolog in the skeletal muscles induces myopathy while elevating its expression muscles prevents statin-induced myotoxicity [104]. Moreover, in humans, mutation of *LPIN1* causes severe rhabdomyolysis [105,106].

Furthermore, *mdy* (*Drosophila* homolog of *DGAT1*) and *Dgat2* (*Drosophila* homolog of *DGAT2*) were significantly reduced in fluvastatin-treated flies (*p* = 0.0056 and 0.0007, respectively) (Appendix A and Table 7). Similarly, muscle biopsy of the statin-treated patient has shown reduced, *DGAT1* and *DGAT2* [107]. The same study suggests that statin treatment reduces the capacity to couple FA and DAG into TG in the skeletal muscles [107]. Taken together, our data suggest that alteration of key enzymes involved in lipid metabolism in muscles or in the whole body may contribute to myopathy, most likely through the accumulation/oversupply of toxic lipid intermediates [104,107,108], which ultimately leads to lipotoxicity.

Lipotoxicity has been shown to compromise insulin signalling in human skeletal muscles [109,110]. We, therefore, looked into the effect of fluvastatin on key genes involved in insulin sensitivity and signalling. *chico* (*Drosophila* homolog of *IRS1*), which encodes the only insulin receptor substrate that functions in an insulin/insulin-like growth factor (IGF) signalling pathway [111,112], was very significantly reduced in fluvastatin-treated flies (*p* = 0.0003) (Appendix A and Table 7). Furthermore, *Thor*, which encodes a eukaryotic translation initiation factor 4E binding protein and is implicated in insulin resistance [113,114], was remarkably reduced (0.0019) (Appendix A and Table 7). *Thor* is also essential for mitochondrial function and synaptic homeostasis in *Drosophila* neuromuscular junctions [114,115]. In total, our data suggest that chronic fluvastatin treatment may disrupt insulin signalling as well.

**Table 7 cells-11-03528-t007:** Effects of chronic fluvastatin treatment on the transcription of important genes involved in lipid metabolism and insulin haemostasis.

	Drosophila Gene/ Protein Name	Statin Effect	Human Homolog	Function
1	*Bmm*/brummer	Upregulated Mean fold change (1.58)	*PNPLA2*/Patatin-like phospholipase domain-containing protein 2 [116,117]	Triglyceride lipase in adipose tissue and muscles [116,117]
2	*Lip1*/Lipase 1	Downregulated Mean fold change (0.64)	*LIPF*/Gastric triacylglycerol lipase [116,117]	Triglyceride lipase and expressed in the digestive system, adipose tissue and muscles [116,117]
3	*Lpin* (Lipin)/ Phosphatidate phosphatase	Downregulated Mean fold change (0.55)	*LPIN1*/Phosphatidate phosphatase LPIN3 [116,117]	Conversion of phosphatidic acid to diacylglycerol during triglyceride biosynthesis and required for insulin signalling. Expressed in many tissues including adipose tissue and muscles [116,117]
4	*mdy* (midway)/ Diacylglycerol O-acyltransferase	Downregulated Mean fold change (0.74)	*DGAT1*/Diacylglycerol O-acyltransferase 1 [116,117]	Triglyceride biosynthesis in the muscles [116,117]
5	*Dgat2*/ Diacylglycerol O-acyltransferase 2	Downregulated Mean fold change (0.43)	*DGAT2*/Diacylglycerol O-acyltransferase 2 [116,117]	Triglyceride biosynthesis and expressed in different tissues [116,117]
6	*CG1941*/ Diacylglycerol O-acyltransferase	No change	*DGAT2*/Diacylglycerol O-acyltransferase 2 [116,117]	Triglyceride biosynthesis [116,117]
7	*CG1946*/ Diacylglycerol O-acyltransferase	No change	*DGAT2*/Diacylglycerol O-acyltransferase 2 [116,117]	Triglyceride biosynthesis [116,117]
8	*Hnf4*	Downregulated Mean fold change (0.43)	*HNF4G*/Hepatocyte nuclear factor 4-gamma [116,117]	A transcription factor involves in fatty acid oxidation, and lipid metabolism [118,119]
9	*Chico*/ encodes a substrate insulin receptor.	Downregulated Mean fold change (0.58)	*IRS1*/Insulin receptor substrate 1 [116,117]	Important for insulin signalling and action [116,117]
10	*Thor*/ encodes a eukaryotic translation initiation factor 4E binding protein	Downregulated Mean fold change (0.51)	*EIF4EBP2*/eukaryotic translation initiation factor 4E binding protein 2 [116,117]	Involved in insulin sensitivity [116,117]

The above-mentioned effects might be systemic or specific in the muscles since the whole body of the flies was used in the qPCR. More studies elucidating specific statin effects on key regulators of lipid metabolism specifically in the skeletal muscles are highly warranted.

### 3.4. Loss of Hmgcr in Skeletal Muscles Results in Phenotypic Changes in Mitochondria and Reduced General Locomotion Activities without Affecting Myofibril Integrity or Climbing Moves and Speeds

Second, we explored possible underlying mechanisms for statin-induced myopathy phenotypes. HMGCR is the main target of statins and many reports have linked SIM to its inhibitory effect on skeletal HMGCR [30,120]. Consequently, we asked whether inhibition of Hmgcr in the skeletal muscles induces locomotion defects and myopathy-like phenotypes similar to fluvastatin treatment. To answer this question, we assessed the impact of Hmgcr knockdown in muscles using the Mhc-Gal4 driver line, as it is specifically expressed in skeletal muscles including the embryonic/larval muscular system and adult flight, legs and labial muscles [121,122,123,124] and it is commonly used in muscles studies [125,126,127]. The knockdown efficiency of *Hmgcr* was confirmed by qPCR (Appendix A). In contrast to both controls, *Hmgcr^RNAi^* knockdown flies (*Mhc-Gal4>UAS-Hmgcr RNAi*) exhibited significantly reduced general locomotion (activity counts/30 min. and total activity counts) as well as the activity while awake (Figure 4A and Appendix A). Surprisingly, in terms of climbing measures, the average position of *Hmgcr^RNAi^* knockdown flies was higher compared to both controls, while the average moves and speed were not significantly changed compared to both groups (Figure 4(B1–B3)). Unexpectedly, at TEM, the dissected leg muscles of *Hmgcr^RNAi^* knockdown flies demonstrated normal myofibril with rare Z-line streaming and the sarcomere length of *Hmgcr^RNAi^* knockdown flies was found to be higher than one control group, *Mhc-Gal4>W1118*, but also lower than the another one, *W1118>UAS Hmgcr^RNAi^* (Figure 4C,D). Vacuolization was present in both *Hmgcr^RNAi^* knockdown flies and one control group, *Mhc-Gal4>w1118*, but more conspicuous in *Hmgcr^RNAi^* knockdown flies (Figure 4C). Strikingly, at the mitochondrial level, *Hmgcr^RNAi^* knockdown flies demonstrated severely fragmented and copious round-liked shaped mitochondria compared to both controls (Figure 4C,F), which may indicate mitochondria dysfunction. However, the mitochondria area of *Hmgcr^RNAi^* knockdown flies was significantly reduced compared only to one control group, *Mhc-Gal4>w1118* (Figure 4E). When compared to fluvastatin-treated CSORC flies, the muscles of *Hmgcr^RNAi^* knockdown flies showed smaller, but similar round-liked shaped mitochondria with more vacuolization and a few occasions of myofibril changes including Z-line steaming (Figure 4G). Since the sarcomere length and myofibril ultrastructure of the *Hmgcr^RNAi^* knockdown flies were not significantly changed compared to both control groups, another mechanism for statin-induced myopathy phenotypes is likely to be involved.

### 3.5. Chronic Fluvastatin Treatment Is Associated with Reduced Expression of the Skeletal Muscle Chloride Channel, ClC-a

Emerging shreds of evidence have indicated that statins inhibit the skeletal muscle chloride channel, CLC-1 in humans and rats [43,44,45,46]. We, thus, hypothesized that fluvastatin may induce myopathy-like phenotypes in flies through inhibition of the skeletal muscle chloride channel as well. To test this hypothesis, first, we looked for a *CLCN1* homolog in *Drosophila*. According to the literature and evolutionary data from PhylomeDB, ClC-a is the closest homolog to *CLCN1* with *a* weighted score, which is obtained by 15 integrative ortholog prediction tools, of 10/15 [117,128,129]. In addition, *CLCN1* is expressed in the muscle fibres and responsible for the Cl− conductance during muscle contraction [128,130]. Therefore, we fed CSORC flies with 1 mM fluvastatin for 5 days and then examined the effect of 1 mM fluvastatin on the ClC-a channel. We dissected leg muscles and fixed them for immunogold labelling using rabbit anti-ClC-2 antibody and then counted the immunogold particles of ClC-a channels. Intriguingly, compared to the control group, fluvastatin-treated flies had a significantly lower number of ClC-a gold particles (*p* = 0.0004) (Figure 5A,B). To verify this result, we performed a Western blot analysis to study the protein expression of ClC-a. Similarly, 1 mM fluvastatin-treated flies displayed a significantly reduced protein expression of ClC-a more than 0.5 mM (Figure 5C and Appendix A). As we mentioned previously, compared to 0.5 mM, 1 mM fluvastatin-treated flies showed a more significant decline in the average speed and position (Figure 2B,C), which might be attributed to the significantly lowered protein expression of ClC-a induced by 1 mM. Surprisingly, when we measured the transcripts of the ClC-a gene, no change in the transcriptional expression was observed in the fluvastatin-fed flies compared to the control (Appendix A), suggesting that the effect of the fluvastatin on the ClC-a is mediated post-transcriptionally, perhaps, on channel proteostasis. However, when we exercised 1 mM fluvastatin-treated flies in the Forced-Climbing System for six hours, the number of ClC-a gold particles was not altered in comparison with the control group (Appendix A), suggesting that the exercise improves the expression of the ClC-a in the skeletal muscles. Consistent with this is the observation that the muscle biopsy from patients and rats treated with fluvastatin exhibit manifestations of myopathy in which the skeletal muscles show lowered ClC-1 protein expression and decreased chloride conductance with no change in *ClCN1* transcript levels [44,45,46]. Additionally, in humans, moderate exercise has been shown to improve the manifestation of SIM, although the data are conflicting [72,131,132]. Hence, future studies investigating the role of the skeletal muscle chloride channel in the exercise-related effects in patients with SIM are highly recommended.

To support this notion, we attempted to mimic the inhibitory effect of fluvastatin on ClC-a by treating flies with two chloride channel blockers, 2-(4-chlorophenoxy) propionic acid (CPP) and 9-anthracene carboxylic acid (A9C), both for five days. Similar to fluvastatin, but to a lesser extent, both chloride channel blockers displayed notably lowered activity counts/30 min and the average activity (CPP *p* ≤ 0.05 and A9C *p* ≤ 0.001) (Appendix A) but the activity while awake was only significantly lower with A9C group (*p* ≤ 0.045, Appendix A). It is worth mentioning that treating skeletal muscle with ClC-1 blockers has been shown to reduce channel activity and cause myotonia in vitro [46,133].

### 3.6. Loss of ClC-a in Skeletal Muscles Induces Myofibril Phenotypic Changes and Impairs the Climbing Ability

Nevertheless, at this point, we did not know whether statin-mediated inhibition of ClC-a induces myopathy-like phenotypes. To address this, we examined the effects of the loss of skeletal muscle ClC-a using the Mhc-Gal4 driver, as it is expressed in skeletal muscles including leg, flight and labial muscles [121,122,123,125,126], and then looked into the myopathy-related phenotypes. The ClC-a knockdown efficiency was confirmed by qPCR (Appendix A). Intriguingly, dissected legs of *ClC-a^RNAi^* knockdown flies (*Mhc-Gal4>ClC-a RNAi*) displayed severe muscle fibre damage including aberrant sarcomere architecture, distorted myofibrils and Z-line streaming along with severely fragmented mitochondria of different shapes and sizes, in comparison to both controls (Figure 6C). Vacuolization was more abundant in *ClC-a^RNAi^* knockdown flies than in both controls. Strikingly, similar to fluvastatin-treated flies, *ClC-a^RNAi^* knockdown flies exhibited noticeably longer sarcomeres and, in a few micrographs, a comparable pattern of myofibril damage including rare occasions of myelin figures (Figure 6F,G). However, in Fluvastatin-treated flies, the myelin figures were bigger and thicker than those in *ClC-a^RNAi^* knockdown flies. Moreover, in contrast to fluvastatin-treated flies, we observed a very rare occasion of round-like-shaped mitochondria in *ClC-a^RNAi^* knockdown flies and the mitochondria area for *ClC-a^RNAi^* knockdown flies was not altered compared to both controls (Figure 6C,F). Furthermore, in terms of climbing ability, similar to the fluvastatin treatment, the average move declined in *ClC-a^RNAi^* knockdown flies compared to both controls (Figure 6(B1)) while the average speed and position were not significantly changed compared to both controls (Figure 6(B2,B3)). Additionally, the activity while wake for the *ClC-a^RNAi^* knockdown flies was also significantly reduced compared to controls, indicating locomotion defects (Figure 6A,B and Appendix A). Astonishingly, the general locomotion activity (activity counts/30 min and total activity counts) was not statistically significantly reduced (Figure 6A).

### 3.7. Chronic Fluvastatin Treatment Is Associated with an Upregulation of Pkcdelta and in Contrast to Fluvastatin, Inhibition of Skeletal Muscle Pkcdelta Improves General Locomotion Activities and Climbing Ability

Against the aforementioned data, a pressing question to pose is how fluvastatin inhibits ClC-a in flies. In humans and rats, PKCθ is a negative regulator of CLC-1, which upon activation, phosphorylates and inhibits CLC1 [43,44,134]. We, therefore, looked for a PKCθ homolog in flies. *D. melanogaster Pkcdelta* alignment with human *PRKCQ*, which encodes PKCθ, gives identity 186/328 (57%) and similarity 244/328 (74%) as well as a weighted score, which is obtained by 15 integrative ortholog prediction tools, of 10/15 [117,135]. One of these 15 integrative ortholog prediction tools is the PhylomeDB which indicates, according to its evolutionary data, that the *Pkcdelta* is the closest homolog to *PRKCQ* [129]. Accordingly, we hypothesized that statins mediate inhibition through the activation of *Pkcdelta*. To test this hypothesis, we fed CSORC flies with 1.0 mM fluvastatin for 5 days and examined the expression of *Pkcdelta*. As expected, fluvastatin-treated flies showed a significant upregulated *Pkcdelta* (*p* = 0.0025, Appendix A). Next, since *Pkcdelta* is highly expressed in the muscles [136,137] and is dispensable for the development and survival of flies [138], we decided to explore the impact of *Pkcdelta* loss on the adult skeletal muscles using the *EDTP-Gal4* driver (also known as *DJ694-Gal4*). EDTP-Gal4 is expressed in the skeletal muscles including legs, labial, flight and abdominal muscles and its expression is restricted solely to the adult stage [125,126,139], making it a specific driver to knockdown *Pkcdelta* in adult skeletal muscles. We confirmed the *Pkcdelta* knockdown efficiency by qPCR (Appendix A). Strikingly, in contrast to both control groups, *Pkcdelta^RNAi^* knockdown flies (*EDTP-Gal4>UAS-Pkcdelta RNAi)* exhibited significantly improved climbing activity (the average moves and speed), general locomotion activities (activity count/30 min and total activity) as well as the activity while awake (Figure 7A,B and Appendix A). These results, except for the activity while wake and the position, showed that loss of *Pkcdelta* in the skeletal muscles exhibited improved locomotion and climbing activity that are in contrast to fluvastatin-treated flies. This may support the notion that fluvastatin may activate Pkcdelta in flies, which may, in turn, inhibit ClC-a and induce lowered locomotion and climbing activities.

It should be noted that PKCθ suppression has been shown to improve myogenesis in mouse myoblasts [103], enhance healing and regeneration of the muscles in a mouse model of muscular dystrophy [140,141] and protect mice from immobilization-induced muscle atrophy [142].

### 3.8. Chelerythrine Chloride, CC, a PKC Inhibitor, Rescues the Fluvastatin-Induced Lowered Locomotion and Climbing Ability

To further confirm the notion that fluvastatin mediated inhibition of ClC-a through activation of Pkcdelta, we attempted to rescue the fluvastatin-induced lowered general activities and climbing ability phenotypes using chelerythrine chloride (CC), a potent PKC inhibitor [143]. First, we treated CSOCR flies with 1 mM fluvastatin for five days and then with CC for three hours. After that, we examined their climbing ability. Intriguingly, CC significantly rescued the fluvastatin-induced lower climbing ability including both average moves and positions. (Figure 8A). Next, we examined the chronic exposure of flies to both fluvastatin and CC for five days in DAMS. CC partially, but significantly, improved the fluvastatin-induced lowered activity counts/30 min only while the total activity was not significantly improved (Figure 8B and Appendix A).

## 4. Discussion

Statin medications are the cornerstone for the prevention and treatment of cardiovascular events [144]. Nevertheless, SIM-associated adverse effects hamper patient compliance and subsequently increase the risk for CVD [8,9,10,11]. Despite extensive studies, the underlying mechanisms for SIM are still poorly understood. Therefore, we employ both pharmacological and genetic approaches to investigate and understand possible underlying causes of SIM using *Drosophila melanogaster.*

We demonstrate, for the first time, that chronic fluvastatin treatment induces morphological damage in myofibrils in *Drosophila* including longer sarcomeres and Z-line losses/streaming (See Figure 3), which are reminiscent of myopathy, in concert with mitochondrial morphological changes, including fragmented round-like shaped and large-sized mitochondria (See Figure 3). Additionally, chronic fluvastatin treatment is accompanied by a remarkable reduction in key genes involved in muscle regeneration and differentiation (See Appendix A) and impaired lipid metabolism and insulin signalling (See Appendix A). Consequently, the general locomotion activities and climbing ability of the flies treated with fluvastatin have been significantly reduced (See Figure 2). Taken together, these findings demonstrate that *D. melanogaster* provides a powerful model system to investigate the SIM phenotypes at multifarious levels.

Mechanistically, we mimicked the statin’s inhibitory effect on Hmgcr by selectively knocking down *Hmgcr* in the skeletal muscles. We found that loss of muscular *Hmgcr* was not sufficient to induce muscular phenotypes (myofibril damage or altered sarcomere size) or impair climbing ability in comparison to fluvastatin treatment (See Figure 2A–C, Figure 3A–D and Figure 4B–D,G). In this context, a single study by Yoshinori et al. has reported that skeletal muscle-specific knockout of *HMGCR* in mice displayed myopathy-like phenotypes that were associated with vacuolated myofibers and necrosis [145]. Nevertheless, it is difficult to relate statin-induced muscle phenotypes to a complete deletion of HMGCR in the skeletal muscles since statins inhibit and diminish, rather than abolish, HMGCR enzyme activity [146]. Moreover, knockdown of zebrafish *HMGCR* (*hmgcrb*) or treating zebrafish embryos with statins has been shown to cause strong myofiber damage with shorter myotube diameters [147], but the authors did not investigate the effect of *hmgcrb* knockdown during the adult stage where the muscles are fully developed. This is important since the mevalonate pathway is indispensable for musculoskeletal development in zebrafish embryos [148].

In flies, the mevalonate pathway, in which Hmgcr is the rate-limiting enzyme, is crucial for isoprenoid biosynthesis, including ubiquinone, prenylated proteins, heme A (Cytochrome C oxidase) and dolichol, in the muscles [149]. In humans, ubiquinone, also called CoQ10, is a crucial node in the electron transfer from mitochondrial Complexes I and II to Complex III in the respiratory transport chains, especially during lipid metabolisms and oxidative stress [150,151]. Consequently, CoQ10deficiency leads to ROS overproduction and impaired FAO [152,153] and ultimately elicits mitochondria dysfunction [154,155,156]. In this context, our data show that knockdown of muscular *Hmgcr* resulted in reduced general locomotion activities along with phenotypic changes in mitochondria including round-shape mitochondria, that were, to some extent, mimicked by fluvastatin treatment (See Figure 3A,B,F and Figure 4A,C,F,G). In addition, fluvastatin treatment has been associated with an upregulation of AMPKalpha (*Drosophila* homolog of AMPK) (See Appendix A). In humans, muscle biopsy from fluvastatin-treated patients displays a high expression of active AMPK, suggesting a cytoprotective response to oxidative stress [45]. Accordingly, it might be reasonable to propose that the reduced general locomotion activities and mitochondria changes associated with fluvastatin treatment are attributed to CoQ10 depletion and subsequent oxidative stress as a result of statin-inhibition of muscular Hmgcr. Nevertheless, as mentioned previously, the data regarding the role of reduced muscular CoQ10 in SIM is conflicting [27,28,30,31,32,33,34] and above all, whether mitochondrial dysfunction should be considered a frequent hallmark of SIM is still disputed [157,158].

We also demonstrate that chronic fluvastatin treatment has been associated with a reduction in the ClC-a expression (See Figure 5) and loss of ClC-a in skeletal muscles has produced myopathy-like phenotypes similar but stronger, to fluvastatin treatment including myofibril ultrastructure damage and longer sarcomere, as well as impaired climbing activity, in particular, the average moves (See Figure 6B–D,F,G). Moreover, exercising flies for six hours restored the altered sarcomere length and normalized chloride channel expression associated with fluvastatin treatment (See Figure 3D, Figure 5B and Appendix A), suggesting that exercise may improve the sarcomere length by enhancing the skeletal muscle chloride channel expression. Collectively, the muscular phenotypes (increased sarcomere length and myofibrillar damage) associated with fluvastatin treatment are perhaps a result of the reduction in the skeletal muscle chloride channel ClC-a. This underscores the significance of the skeletal muscle chloride channel in the histopathology of SIM, which is backed up by a recent report which has identified truncating mutations in *CLCN1* (rs55960271) in patients with SIM [159]. These mutations are also the most frequent pathogenetic mutations in myotonia [159]. It is noteworthy that mutation or loss of function of CLC-1 leads to decreased resting conductance and potassium accumulation [160,161]. This, in turn, elicits hyper-excitability, muscle stiffness, and myotonia symptoms [48,160,161,162,163,164,165]. Furthermore, in humans, moderate exercise has been shown to improve the manifestation of SIM, although the data are conflicting [72,131,132]. Therefore, future studies investigating the role of the skeletal muscle chloride channel in the exercise-related effects in patients with SIM are highly recommended.

It has been reported that through the activation of protein kinase c theta (PKCθ), statins, including fluvastatin, reduce the expression of a skeletal muscle chloride channel, chloride channel 1 (CLC-1), and thereby the chloride resting potential (gCl) [43,44,45,46]. As mentioned earlier, PKCθ is a negative modulator of CLC-1 in humans and rats [43,44,134].

Along the same lines, our data show that fluvastatin treatment has been associated with an upregulation of *Pkcdelta* and, contrary to fluvastatin, loss of *Pkcdelta* in the skeletal muscles has led to improved general locomotion activity and climbing measures (See Figure 2 and Figure 7). This leads us to postulate that the statin inhibition ClC-a may be mediated by the activation of Pkcdelta. This possibility is supported by the fact that the locomotion and climbing phenotypes associated with fluvastatin treatment have been partially but significantly rescued by chelerythrine chloride (CC), a potent PKC inhibitor [143], (See Figure 8). It is noteworthy that CC has also been shown to partially restore the declined resting chloride channel conductance (gCl) on the dissected muscle of fluvastatin-treated rats [44] and improves the contraction of the injured skeletal muscle fibres of the rat [166].

So far, evidence for the mechanisms underlying statin-activation of PKCθ in humans is still lacking. It has been suggested that statins induce the mitochondrial release of Ca^2+^ which then activates PKCs [45,46,134]. However, PKCθ, and Pkcdelta, belong to novel PKCs (nPKCs) which are insensitive to Ca^2+^ because their C2 domain lacks the residues for Ca^2+^ binding [167,168]. This could prompt the possibility that statins could mediate the activation of PKCθ through ROS or lipid intermediates, such as FFA and DAG. Regarding ROS, in contrast to fluvastatin-treated flies and the *ClC-a^RNAi^* knockdown flies, loss of Hmgcr (and maybe the consequent ROS generation) was not enough to elicit myofibril ultrastructure or alter the sarcomere length. Further studies are needed to better understand the role of Hmgcr inhibition or the subsequent decline of its downstream products in the statin-mediated inhibition of ClC-a.

Previous studies have suggested impaired lipid metabolism in patients with SIM [107,110,169]. Furthermore, lipid intermediates, specifically DAG and FFA, have been reported to activate PKC-θ in humans, both, directly and by enhancing its membrane translocation, respectively [170,171,172,173,174]. Consistent with this possibility, in fluvastatin-treated flies, genes regulating TG metabolism have been largely altered (See Appendix A and Table 7), which may result in the accumulation of such toxic lipid intermediates. Since FFA and DAG can also activate other forms of nPKCs [170,171,172,173,174], we examined the effect of the fluvastatin treatment on the expression of *Pkc98E* (*Drosophila* homolog of PRKCE which encodes PKCε), which belongs to the nPKCs and is also expressed in muscles [117,137]. Expectedly, chronic fluvastatin treatment was also associated with an upregulation of *Pkc98E* (*p* = 0.0077, Appendix A). It is therefore reasonable to propose that fluvastatin-induced activation of Pkcdelta could be mediated by accumulated lipid intermediates such as FFA or DAG. This possibility is backed up by data showing that patients with SIM have an elevation of FFA and DAG levels, and activation of PKC and, by the same token, pharmacological inhibition of PKC reverses and restores the statin-induced insulin in the skeletal muscles [107,175]. In this context, in addition to its inhibitory effect on PKC, CC has demonstrated anti-inflammatory and antioxidant effects [176,177] and improved lipid metabolism, particularly fatty acid oxidation, and insulin haemostasis in mice [178] which could suggest additional mechanisms for its rescuing effect on fluvastatin-induced lowered locomotion and climbing phenotypes (See Figure 8). Although impaired lipid metabolism has been reported in SIM [104,107], the underlying mechanisms remain to be elucidated.

Interestingly, our study suggests that depletion of sarcolemma cholesterol is not the main underlying cause of SIM because fruit flies cannot biosynthesize cholesterol de novo and instead obtain it from the food [149,179]. This may exclude the possibility that the locomotion and myopathy phenotypes associated with fluvastatin are caused simply by cholesterol depletion. This possibility is backed up by the fact that, in humans, patients with hereditary cholesterol biosynthetic defects do not show clinical manifestations of skeletal myopathy [120].

A strength of our study is that we have used both genetic and pharmacological approaches to decipher the phenotypes associated with statins treatment using an affordable and easily genetically manipulated platform system, *Drosophila melanogaster*. In addition, we were able to confirm the knockdown efficiency for each flies’ crosses, which validated the resulting phenotypes. This study has several limitations. Although we quantified sarcomere lengths using multiple images of the same leg section to robustly detect any differences in sarcomere lengths between control and treated/knockdown flies, we cannot exclude the possibility that different muscle types may have been abundant in the sections that were analysed. This may add to a certain variability in the outcome measurements. Future studies would benefit from examining the differences in the sarcomere lengths within each individual type of femur muscle (tidm, tilm, and tirm) or thorax muscle. In addition, due to the lack of specific antibodies in the market, we could not determine whether fluvastatin activates Pkcdelta which could have been performed with Western blot for its active and inactive form with the appropriate antibodies. Furthermore, we also did not demonstrate whether Pkcdelta regulates (phosphorylates and inhibits) skeletal muscle ClC-a in flies, as it is outside of the scope of this study and its homologue, PKC-θ, has already been shown to inhibit the skeletal muscle ClC-1 in humans and rats [43,44,134]. Although we demonstrate that CC mechanistically rescued the locomotion and climbing phenotypes accompanied by fluvastatin, we did not show if CC rescued the myofibril or mitochondrial phenotypes caused by fluvastatin. The reason behind this is that we do not know exactly when the muscle or mitochondrial phenotypes started to emerge after statin treatment. Moreover, we did not measure ROS in the fluvastatin-treated flies or *Hmgcr* knockdown flies since this has been studied extensively [180,181]. Finally, we did not test several types of statins, which would help to demonstrate the differential effects of lipophilic and hydrophilic statins on the muscles, which is warranted to be addressed in future studies.

## 5. Concluding Remarks

This study demonstrates that statins induce locomotion and muscular phenotypes that are reminiscent of the human myopathy in *Drosophila melanogaster*, suggesting that the mechanisms behind the important side effects of statins could be highly evolutionarily conserved. The statins may induce mitochondria phenotypic changes through direct inhibition of muscular Hmgcr and myofibrillar damage reminiscent of myopathy that is associated with inhibition of skeletal muscle ClC-a. The study also underscores the role of skeletal muscle chloride channels for statin muscular phenotypes and its potential, perhaps, as a biomarker for SIM for future clinical studies. Moreover, the study sheds light on a possible mechanistic pathway of the statin-mediated PKC inhibition of skeletal muscle chloride channels as well as the contribution of the impaired lipid metabolism. This study also underlines the potential of chelerythrine as a pharmacological tool for understanding statin myopathy-related phenotypes for future application. Collectively, our findings demonstrate that *D. melanogaster* provides a powerful and affordable model system to elucidate statin myopathy phenotypes in multifaceted aspects.

## Figures and Tables

**Figure 1 cells-11-03528-f001:**
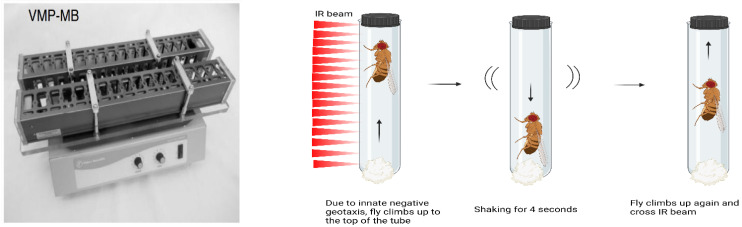
Schematic representation for the Forced-Climbing System. The left image is VMP-MB where the upper portion is two slots of MultiBeam Activity Monitor (MB), each has 16 slots for tubes and each slot is equipped with 17 IR beams and the lower portion is Vortexer Mounting Plate, (VMP) which is used as a shaker. The right picture shows how the Forced-Climbing System works. Simply, an anaesthetized fly was placed inside the tube plugged with cotton from the bottom and with a black cap from the top. Then, the tube was placed inside MB. Each MB consists of 16 slots. The VMP agitates tubes inside MB for 4 s each minute. The fly tends to climb up against gravity due to innate negative geotaxis. Then, after a four-second shaking period, the fly falls to the bottom and climbs up again to the top of the tube and crosses the 17 IR beam. The left image was taken from TriKinetics.com, accessed on 1 August 2022 (with permission) while the right was created with Biorender.com.

**Figure 2 cells-11-03528-f002:**
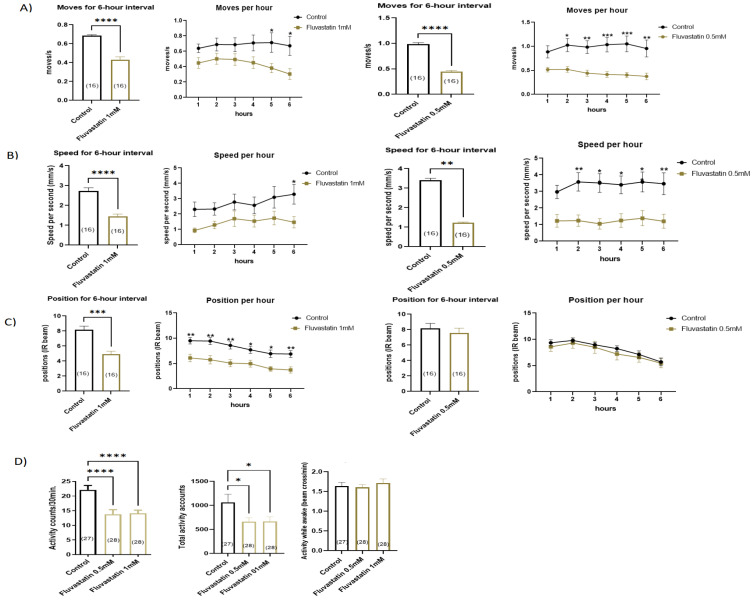
Fluvastatin treatment-induced lower climbing ability and general locomotion activity in CSORC flies. In 0.5 and 1 mM fluvastatin groups, flies were fed with food mixed with a corresponding volume of fluvastatin, while in control groups, flies were fed with food, all for five days before they exercised in the Forced-Climbing System for six hours (**A**) the histograms show the average moves for a combined six-hour interval and per hour, (**B**) the histograms show the average speed for a combined six-hour interval and per hour, and (**C**) the histograms show the average position for a combined six-hour interval and per hour. Using DAMS, flies were treated with 0.5 mM or 1 mM fluvastatin mixed with food in the fluvastatin groups, whereas in the control group, flies were treated with food only, both for five days (**D**) the histograms show the general locomotion activity (activity counts/30 min and total activity) and activity while awake of the flies. The Shapiro–Wilk tests were used to check normality where a parametric test was used for normally distributed data and a non-parametric test for non-normally distributed data. The statistical difference between the three groups was calculated using one-way ANOVA with Bonferroni’s multiple comparisons for normally distributed data and Kruskal–Wallis one-way tests for otherwise data. For two groups, except for moves, speed and position per hour where two-way ANOVA with Bonferroni’s multiple comparisons test was used, Student’s *t*-test or Mann–Whitney test was used to calculate statistical difference according to the normality test whereby * *p* ≤ 0.05; ** *p* ≤ 0.01; *** *p* ≤ 0.005; **** *p* ≤ 0.0001. In each histogram, the bars represent the mean values ± SEM and the number of flies is indicated in brackets.

**Figure 3 cells-11-03528-f003:**
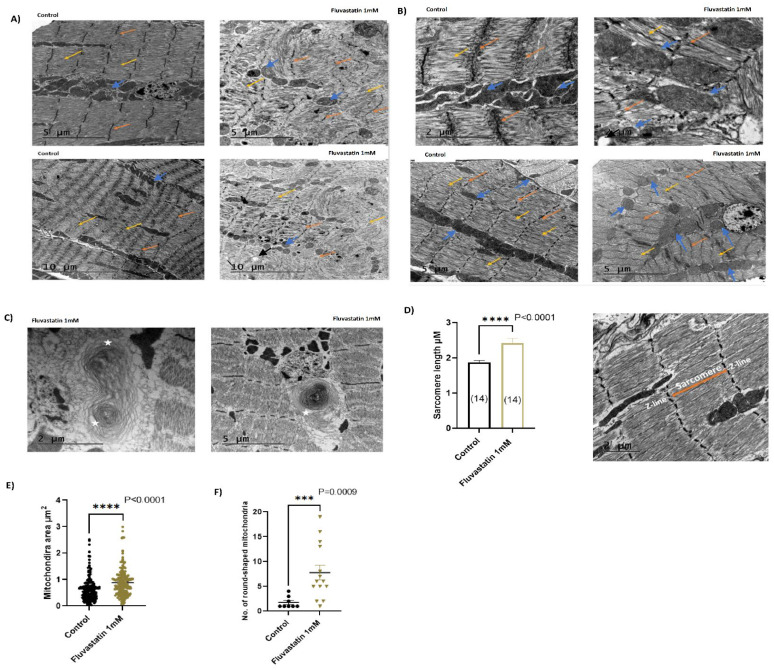
Fluvastatin induces myopathy-like phenotypes in flies. (**A**,**B**) TEM micrographs for the morphology for the longitudinal section of the dissected skeletal muscles of the femur segment of CSORC fly legs, wherein the fluvastatin group, CSORC flies were treated with 1.0 mM fluvastatin while the control group was fed with food, both for five days. Compared to the control group, the muscles of the fluvastatin group showed pronounced abnormal myofibril architecture including distorted sarcomere, Z-line streaming, and very few vacuoles, as well as more fragmented mitochondria with different shapes and sizes. The orange arrow indicates sarcomere, yellow indicates Z line, black indicates vacuoles and blue indicates mitochondria. (**C**) Micrographs show myelinic figures (stars) similar to toxic myopathy in the fluvastatin group. (**D**) Using ImageJ software, we measured the sarcomere length between two adjacent Z-lines. On the left, the histogram shows the sarcomere length in µm which was measured in 14 TEM micrographs for each group (fluvastatin and control). On the right is a representation of the way the sarcomere length was measured. (**E**) Using ImageJ software, we traced the mitochondria area in selected TEM micrographs. The histogram represents the area of mitochondria in µm^2^ where the dots represent the number of mitochondria. (**F**) Using ImageJ software, we counted the number of round-like shaped mitochondria in TEM micrographs where the dots represent the number of mitochondria in each micrograph. In each histogram, the bars represent the mean values ± SEM. The Shapiro–Wilk test was used to check normality. Accordingly, for normally distributed data, Student’s *t*-test was used to calculate statistical significance and Mann–Whitney test for otherwise data whereby *** *p* ≤ 0.005; **** *p* ≤ 0.0001.

**Figure 4 cells-11-03528-f004:**
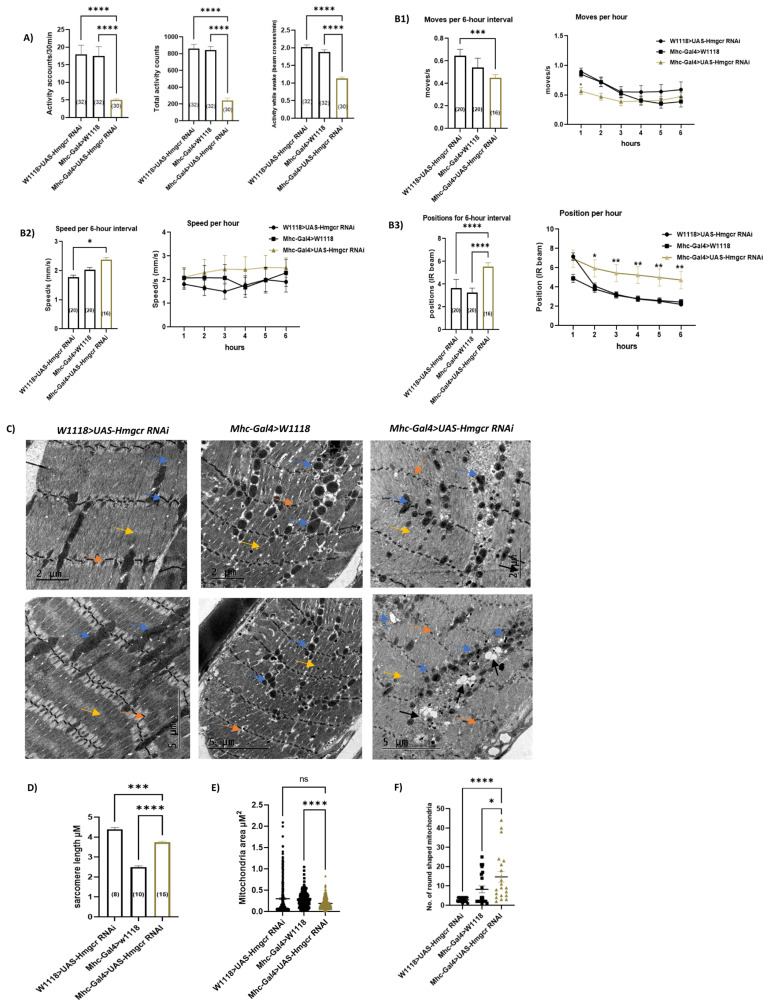
*Hmgcr* knockdown on skeletal muscles results in reduced locomotion activity and mitochondrial phenotypic changes. *Hmgcr* knockdown was carried out by using the Gal4-UAS system, where *Hmgcr^RNAi^* knockdown group is *Mhc-Gal4>UAS-Hmgcr RNAi* and control groups are *Mhc-Gal4>w1118* and *w1118>UAS-Hmgcr RNAi*. Male flies, aged 1-2 days after eclosion and fed with normal food, were collected for experiments. (**A**) The histograms show the general locomotion activity (activity counts/30 min and total activity) and activity while awake using DAMs and the number of flies is indicated in the brackets. (**B1**–**B3**) The left histograms indicate the average moves, speed and positions for a total six-hour interval while the right show the same per hour for flies using the force climbing locomotion system and the number of flies is indicated in the brackets. (**C**) TEM micrographs for the morphology of the longitudinal section of leg skeletal muscles show normal sarcomere (orange arrow) architecture and myofibrils and Z-line (yellow arrow) along with more capacious vacuoles (black arrow) in *Hmgcr^RNAi^* knockdown flies comparable to both controls. The mitochondria (blue arrows) are more ring-shaped and severely fragmented in *Hmgcr^RNAi^* knockdown flies. (**D**) Using ImageJ software, the sarcomere length, in µM, was measured in TEM micrographs (indicated in brackets) in each group. (**E**) Using ImageJ software, we traced the mitochondria area in TEM micrographs. The histogram represents the area of mitochondria in µm^2^ where the dots represent the number of mitochondria. (**F**) Using ImageJ software, we counted the number of round-like shaped mitochondria in TEM micrographs where the dots represent the total number of mitochondria in each TEM micrograph. (**G**) TEM micrographs for the morphology of longitudinal sections of leg skeletal muscles for *Hmgcr^RNAi^* knockdown flies and CSORC flies treated with 1 mM fluvastatin for five days. Each bar represents the mean value (±SEM). For B 1, 2 and 3, two-way ANOVA with Bonferroni’s multiple comparisons was used. For (**A**,**D**–**F**), first, the Shapiro–Wilk test was used to check normality; accordingly, the statistical difference was calculated using one-way ANOVA with Bonferroni’s multiple comparisons for normally distributed data and Kruskal–Wallis one-way tests for otherwise data whereby * *p* ≤ 0.05; ** *p* ≤ 0.01; *** *p* ≤ 0.005; **** *p* ≤ 0.0001.

**Figure 5 cells-11-03528-f005:**
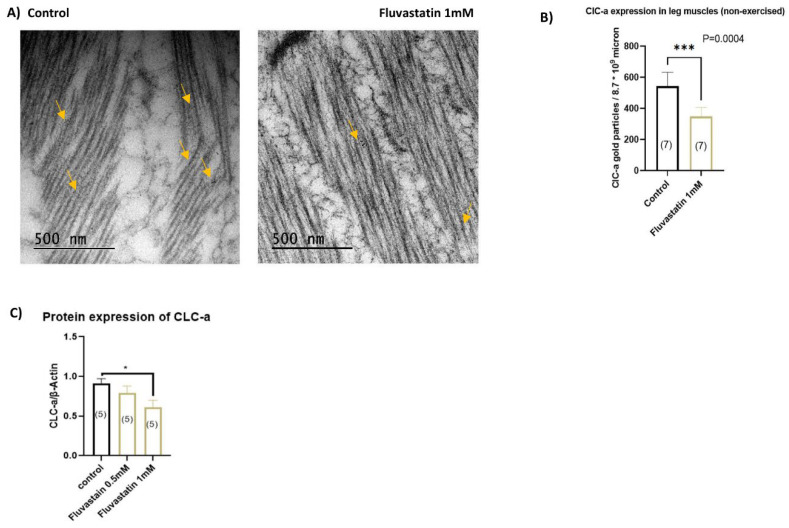
Fluvastatin significantly lowered the expression of the ClC-a channels in the leg’s skeletal muscles of CSORC flies. (**A**) TEM micrographs of double labelling using CLC-2 (rabbit anti-ClC-2 antibody) show immunogold particles of ClC-a (yellow arrow) in the femur segment of the legs of CSOCR flies. The morphology of the muscles is poor due to the treatment necessary for immunolocalization. The skeletal chloride channel, ClC-a, is localized along the T-tubules. (**B**) The histogram represents the quantification of ClC-a gold particles in dissected skeletal muscles of femur and tibia segments of CSORC fly legs using ImageJ software. The Shapiro–Wilk test was used to check normality and accordingly, Student’s *t*-tests were used to calculate the significant difference between groups. The number of TEM grids in each group is indicated in brackets. (**C**) The histogram represents the quantification of the protein relative levels of ClC-a whereby the relative protein intensity was calculated by normalising the absolute intensity of ClC-a protein with β-Actin. The whole body of 10 flies were used per sample and the number of replicates is indicated with the brackets. The Shapiro–Wilk test was used to check normality and accordingly, one-way ANOVA with Bonferroni’s multiple comparisons test was used to calculate statistical significance whereby * *p* ≤ 0.01; *** *p* ≤ 0.005. Each bar represents the mean value (±SEM).

**Figure 6 cells-11-03528-f006:**
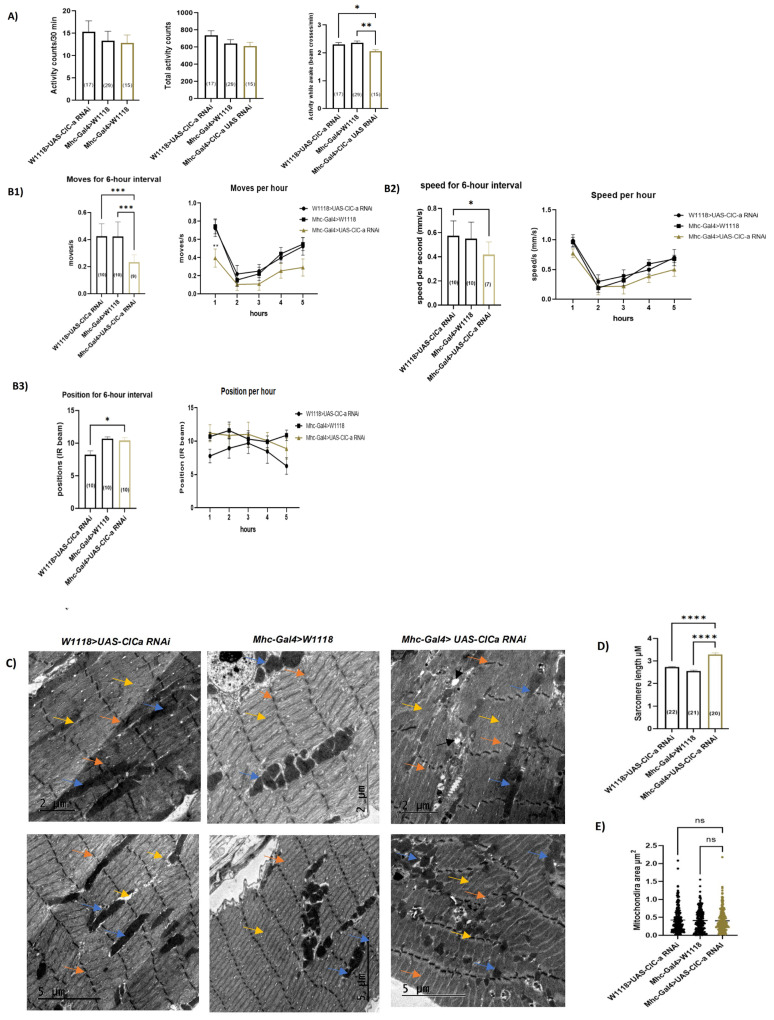
ClC-a knockdown in muscles lowered climbing movement and induced myopathy-like phenotypes in flies. ClC-a knockdown was carried out using the UAS-Gal4 system where *ClC-a^RNAi^* knockdown group is *Mhc-Gal4>UAS-ClC-a RNAi* and control groups are *Mhc-Gal4>w1118* and *w1118>UAS-ClC-a RNAi*. Male flies, aged for five days after eclosion and fed with food, were collected for the experiments. (**A**) The histograms indicate the activity counts/30 min, total activity accounts and activity while awake using DAMs where the number of flies is indicated in brackets. (**B1**–**B3**) The histograms indicate the average moves, speed and positions for a total six-hour interval on the left and per hour on the right using the Force-Climbing System where the number of flies is indicated in brackets. (**C**) TEM micrographs for the morphology of the longitudinal sections of the skeletal muscles of the femur segment of the legs. *ClC-a^RNAi^* knockdown flies show abnormal sarcomere architecture, disoriented, distorted myofibrils and Z-line streaming along with severely fragmented mitochondria of different shapes and sizes in comparison to controls. Orange arrow for Z-line, yellow for sarcomere, black for vacuoles, and blue for mitochondria. (**D**) Using ImageJ software, the sarcomere length, in µM, was measured in a selected number of TEM micrographs for each group (indicated in brackets). (**E**) Using ImageJ software, we traced the mitochondria area in selected TEM micrographs. The histogram represents the area of mitochondria in µm^2^ where the dots represent the number of mitochondria. (**F**) TEM micrographs illustrate a similar pattern of muscle fibre damage between *ClC-a^RNAi^* knockdown flies and CSORC flies fed with 1.0 mM fluvastatin for five days while (**G**) shows the myelin figure (star) in both groups. In each histogram, the bars represent the mean values ± SEM. The number of samples is indicated in brackets. The Shapiro–Wilk test was used to check normality wherein (**A**,**D**,**E**) the statistical difference was calculated using one-way ANOVA with Bonferroni’s multiple comparisons for normally distributed data and Kruskal–Wallis one-way tests for non-parametric data. For (**B1**–**B3**), two-way ANOVA with Bonferroni correction for multiple comparisons test was used. * *p* ≤ 0.05; ** *p* ≤ 0.01; *** *p* ≤ 0.005; **** *p* ≤ 0.0001.

**Figure 7 cells-11-03528-f007:**
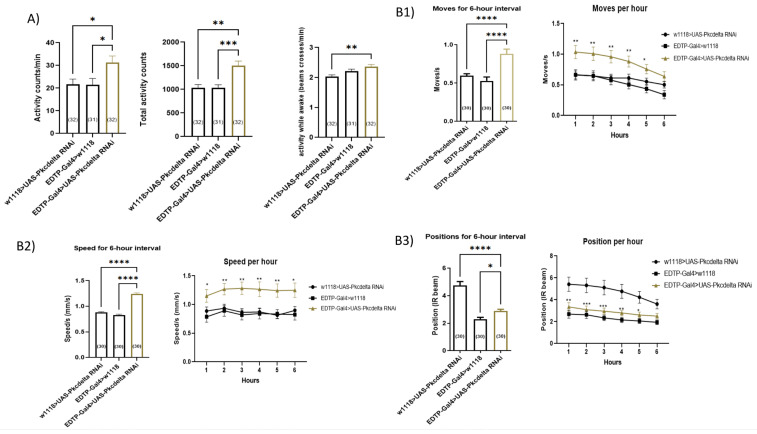
*Pkcdelta* disruption in the skeletal muscles improved general locomotion activity and climbing ability. *Pkcdelta* knockdown was carried out by using the Gal4-UAS system where the *Pkcdelta^RNAi^* knockdown group is *EDTP-Gal4>UAS-Pkcdelta RNAi* and controls are *EDTP-Gal4>w1118* and *w1118>UAS-Pkcdelta RNAi*. Male flies, aged five days after eclosion, were collected for DAMS and LAMS and all flies were fed with normal food for five days. (**A**) The histograms show the activity counts/30 min, the total activity counts and activity while waking for five days using DAMS. (**B1**–**B3**) The left histograms show the average moves, speed and positions for a six-hour interval while the right histograms show the same per hour using the Forced-Climbing System. Each bar represents the mean value (±SEM) and the number of flies is indicated in the brackets. For (**A**), the Shapiro–Wilk test was used to check the normality test; accordingly, the statistical difference was calculated using one-way ANOVA with Bonferroni’s multiple comparisons for normally distributed data and Kruskal–Wallis one-way tests for otherwise data. For (**B1**–**B3**), two-way ANOVA with Bonferroni’s multiple comparisons test was used whereby * *p* ≤ 0.05; ** *p* ≤ 0.01; *** *p* ≤ 0.005; **** *p* ≤ 0.0001.

**Figure 8 cells-11-03528-f008:**
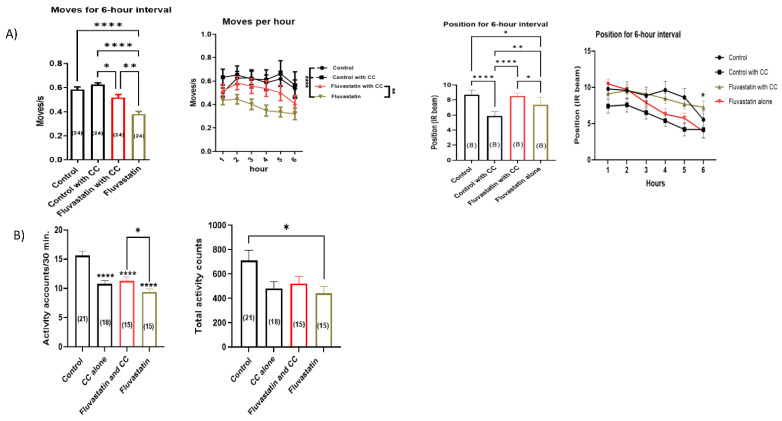
Chelerythrine chloride, CC, partially but significantly rescued the fluvastatin-induced lowered activity counts and climbing ability. Male virgin CSORC flies were used for the experiments. (**A**) The left histograms show the average moves for six-hour intervals and the moves/s per hour while the right histograms show the same for the average position. In the control group, flies were fed with food for five days; in the CC group, flies were fed with normal food for five days and then on the fifth day, they were fed with 100 μM CC for three hours; in fluvastatin with CC group, flies fed with 0.5 mM fluvastatin for five days and then in the fifth day, they fed with 100 μM CC for three hours; and in fluvastatin group, flies fed with 0.5 mM fluvastatin for five days. Then, flies were exercised for six hours using the force-climbing activity system (**B**) The histograms show the activity counts/30 min and total activity counts in comparison to the control. The control group was fed with normal food alone, the fluvastatin group was fed with 0.5 mM fluvastatin, the fluvastatin and CC group was fed with both 0.5 mM fluvastatin and 100 μM CC, and the CC group was fed with 100 μM CC. For (**A**) two-way ANOVA with the Bonferroni test was used to compare between groups. For (**B**) The Shapiro–Wilk test was used to check normality and accordingly, one-way ANOVA with Kruskal–Wallis test was used for comparing all groups with control, and Mann–Whitney’s test was used to calculate the statistical significance between fluvastatin and fluvastatin and CC groups. Each bar represents the mean value (±SEM) and the number of flies is indicated in the brackets where * *p* ≤ 0.05; ** *p* ≤ 0.01; **** *p* ≤ 0.0001.

**Table 1 cells-11-03528-t001:** Fly stock purchased from the Bloomington Stock Center and used in this paper.

Bloomington Stock Numbers	Drosophila Line	Referred to as
50652	y[1] v[1]; P{y[+t7.7]v[+t1.8] = TRiP.HMC03053}attP40	*UAS-Hmgcr RNAi*
8176	w[1118]; P{w[+mW.hs] = GawB}EDTP[DJ694]	*EDTP-Gal4*
28355	y[1] v[1]; P{y[+t7.7] v[+t1.8] = TRiP.JF02991}attP2	*UAS-Pkcdelta RNAi*
53337	P{y[+t7.7] v[+t1.8] = TRiP.HMC03566}attP40/CyO	*UAS-ClC-a RNAi*
38464	w[*]; P{w[+mC] = Mhc-RFP.F3-580}2, P{w[+mC] = Mhc-GAL4.F3-580}2/SM6b	*Mhc-Gal4*

**Table 2 cells-11-03528-t002:** Acute and chronic feeding pattern for fluvastatin groups and control group.

Experimental Group	Solution	Feeding Period	On the Fifth Day
Control	4.5 mL food and 0.5 mL water	five days	Starved flies for one hour and then transfer to 100 μL of sucrose alone for three hours
0.5 mM Fluvastatin (five days)	4.5 mL food and 0.5 mL of 5 mM fluvastatin	five days	Starved flies for one hour and then transfer to 100 μL of sucrose alone for three hours
0.5 mM Fluvastatin (three hours)	4.5 mL food and 0.5 mL of 5 mM fluvastatin	five days	Starved flies for one hour and then transfer to 100 μL of 0.5 mM fluvastatin for three hours

**Table 3 cells-11-03528-t003:** Feeding pattern for fluvastatin group, fluvastatin with CC group, control group, and control with CC group.

Experimental Group	Solution	Feeding Period	On the Fifth Day
Control	4.5 mL food and 0.5 mL water	five days	Starved flies for one hour and then transfer to 100 μL of sucrose alone for three hours
Control with 100 μM CC	4.5 mL food and 0.5 mL water	five days	Starve flies for one hour and then transfer to 100 μL of 100 μM CC for three hours
0.5 mM Fluvastatin	4.5 mL food and 0.5 mL 5 mM fluvastatin	five days	Starved flies for one hour and then transfer to 100 μL of 0.5 mM fluvastatin for three hours
0.5 mM Fluvastatin with 100 μM CC	4.5 mL food and 0.5 mL 5 mM fluvastatin	five days	Starved flies for one hour and then transfer to 100 μL of 100 μM CC for three hours.

**Table 4 cells-11-03528-t004:** Feeding pattern for fluvastatin group, fluvastatin with CC group, CC group and control groups.

Group	Stock	Volume Taken	Total Volume with Food
1.0 mM Fluvastatin	3 mM fluvastatin	4 mL	12 mL
0.5 mM Fluvastatin	3 mM fluvastatin	2 mL	12 mL
0.5 mM Fluvastatin with 100 μM CC	3 mM fluvastatin 2.9 mM CC	2 mL fluvastatin 413 μL CC	12 mL
100 μM CC	2.9 mM CC	413 μL CC	12 mL
Control	no	no	12 mL

**Table 5 cells-11-03528-t005:** Flies feeding pattern for the chloride channel blockers groups and control.

Group	Stock	Volume Taken	Total Volume with Food
Control	no	2.4 mL of alcohol	12 mL
1 mM CPP	5 mM CPP in alcohol	2.4 mL of the stock solution	12 mL
1 mM A9C	5 mM CPP in alcohol	2.4 mL of the stock solution	12 mL

**Table 6 cells-11-03528-t006:** Sequences for the primers used in the study.

Primer Name	Forward Sequence	Reverse Sequence
*Hmgcr*	5′GCTGCACTGCCGTACTGTA3′	5′AATGCCCAGCACATATTTGGA3′
*CLC-a*	5′TGGGCGAGGATTGGGTATTC3′	5′ACTGAACAAAAGGCTGTGACG3′
*Pkc98E*	5′TCCATTATGCACGACGATGT3′	5′CTCCGGATTTTTGGTGAGAA3′
*Pkcdelta*	5′GGCACCAAACACCCGTATCT3′	5′CCCATAGAATCTGGCTCGCT3′
*RPL32*	5′AGCATACAGGCCCAAGATCG3′	5′TGTTGTCGATACCCTTGGGC3′
*AMPK ALPHA*	5′TACCAGGTCATATCGACGCC3′	5′ACGCCAGAGATAATCTGCTGAA3′
*mdy*	5′CACAAAGTGTGACTCGTGCTG3′	5′CCAGTTCACCAGTCCAGAAAAA3′
*Dgat2*	5′CAGATACTGGTCACGGCCTTT3′	5′CGGATTGGGTTTTCTTGTGGTG3′
*Hmgcr*	5′GCTGCACTGCCGTACTGTA3′	5′GCCACAAGCACCAGGATG3′
*Mef2*	5′ATATCACGCATCACCGATGAAC3′	5′GGCGTACTGGTACAGCTTGT3′
*Mhc*	5′CCAAGACGGTCAAAAACGAT3′	5′GATGTTGGCTCCCGAGATAA3′
*zfh1*	5′CCTCCAAGAAGTGCATCAGCA3′	5′CGAAGTAGCTCATCGGATGTG3′
*Hnf4*	5′GGCGACGGGCAAACATTATG3′	5′CGCAAATCTCGCAAGTGTACTGAT3′
*Bmm*	5′GTCTCCTCTGCGATTTGCCAT3′	5′CTG AAG GGA CCC AGG GAG TA3′
*lip1*	5′GTG AGCCTGGCCTACTTGC3′	5′GGTCGAGGGTGGTGTGATTC3′
*Lpin*	5′CACACCGACAACACACTGGA3′	5′CTTCTTCTCGCCCTGAAACAG3′
*CG1941*	5′GTCTACGCGAATCACAAGAGAA3′	5′CGATAATGCCGAAACAGCCAA3′
*CG1946*	5′CAGACCTGGTACGTCATTC3′	5′CGCCGTAGTACAGCAGGATAG3′
*Chico*	5′GCGCACTCACCTTATGACCA3′	5′GCACACGAATGTCAGGGATTT3′
*Thor*	5′CGTCCAGCGGAAAGTTTTCG3′	5′GTTTGGTGCCTCCAGGAGTGG3′

## Data Availability

Data are contained within the article and Appendix A.

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
