# Peer review of "Statins Induce Locomotion and Muscular Phenotypes in Drosophila melanogaster That Are Reminiscent of Human Myopathy: Evidence for the Role of the Chloride Channel Inhibition in the Muscular Phenotypes"

_cells, 2022, doi:10.3390/cells11223528_

Round 1

Reviewer 1 Report

This is a very interetsing manuscript with extensive experientation used ot support the DM model system in SIM.  My comments are relatively minor, but I feel may strengthen the paper overall.

1) There is no hypothesis stated in the Abstract  or Introduction.  Although the reader may infer the hypothesis, the Authors should inlcude a specific statement of the hupothesis tested in the study.

2) The cause of the increase in sarcomere length is not addressed, nor is the effect of longer sarcomeres on overall muscle function in this system.  In the DM muscles with longer sarcomeres, is there longer actin and myosin filaments, or is there greater spacing in the I-Band or A-Band regions?  It would seem that greater spacing in the I-Band and/or A-Bands would lead to differently functioning muscles, but the Authors should include a sentence or two on their thoughts on this topic.

3) Presenting larger EM images to support the larger sarcomeres should be considered, along with assessments of actin and myosin lengths.

Author Response

Thank you for your valuable comments. We have made some adjustments accordingly. Please, find our reply to your comments. 

Reviewer 2 Report

This is an interesting study on the effects of a statin on muscle structure and function in the Drosophila system. Electron microscopy is utilized to look at structure. A number of assessments of muscle function in adults are provided. Further, the manuscript provides evidence for chloride channel involvement in the statin mechanism. There are quite a number of concerns regarding the experimental data, most notably the use of leg muscles of unclear origin as well as the interpretation of the morphology. The paper is fairly well written, but I provide a number of important suggestions for textual improvement. While this is a potentially important and impactful study with possible clinical relevance, a number of strong concerns need to be addressed before it would be suitable for publication. 

MAJOR CONCERNS:

1. The authors provide no discussion of how the dosage of Fluvastatin correlates with dosages given to human patients. Is it similar? Many times higher? The dosage level compared to the human therapeutic dose is important to discuss, since many “non-toxic” compounds will have severe effects when extraordinary dosages are utilized. 

2. Drosophila legs contain multiple types of muscle. The legs are composed of coxa, trochanter, femur, tibia, 5 tarsal segments and claw (https://www.jove.com/t/62844/dissection-immunohistochemistry-drosophila-adult-leg-to-detect). Muscles within each section are presumably morphologically different (see table 1 and figure 6 in the following publication: https://journals.biologists.com/dev/article/131/24/6041/42694/Coordinated-development-of-muscles-and-tendons-of ). Aside from being “cut at the femur”, there is no mention of which specific segment and muscle(s) were observed. This is most critical in interpreting the electron micrographs in Figures 3 and 4, as well as supplementary Figure 2A (certainly the bottom two panels are not the same muscle type), as the differing morphologies observed may arise from looking at different muscle types, rather than as a result of statin treatment. The authors need to ascertain whether they were indeed observing the same leg muscle type in all samples and explain which muscle they studied. This applies to all the micrographs in the paper as well as the sarcomere length measurements and mitochondrial data. 

3. Figure 3: unclear which are transverse and which are longitudinal sections. Why do the control samples have colored arrows indicating distorted sarcomeres, Z-line streaming, etc.?  

4. Were the muscles treated to ensure they were relaxed? For instance, in Figure 4C, there are wide I-bands in some samples and narrow I-bands in others. This could be because they are different muscle types or are differentially contracted. 

5. Are the authors confident that clear areas are vacuoles, as opposed to parts of the tracheal system? 

6. All conclusions based upon the qRT-PCR data need to be tempered as to the effects observed being relevant to muscle, since whole animals were used for RNA isolation and the knockdown effects (particularly for genes that are expressed in non-muscle tissues) may not be muscle-specific. In fact, they may not be relevant to muscle expression at all. 

7. For Table 7, it would be helpful to list the fold up or down regulation.

8. Section 3.4. The Mhc promoter is also expressed in embryonic and larval muscles. Perhaps this has no effect on the adult phenotypes, but it should be mentioned.

9. For figure 4, I do not understand the text stating that “the average moves and speed of HmgcrRNAi knockdown flies 560 were comparable to both controls (Figure 4 B1, 2 & 3).” I see statistical differences in all of the cited bar graphs. 

10. For figure 5, again, are we looking at the same leg muscles? Which ones? The morphology is poor, presumably due to the treatment necessary for immunolocalization. Where is the channel supposed to be localized? To the T-tubules? Is this consistent with the observed gold particles? 

11. Figure 6C purports to depict transverse sections. The resolution is not adequate to see thick and thin filaments in cross-section. However, certainly for some of the panels, it appears that Z-lines are present and the sarcomeres are actually shown in longitudinal section. In fact, it appears that the controls are in longitudinal section and the RNAi samples are in transverse section, i.e., these cannot readily be compared. Likewise, it appears that different muscle types are being compared. For instance, in the top panels of the controls for section D, the sarcomere lengths are extraordinarily different. These are major concerns in regard to interpreting the structural data for all the figures. 

MINOR CONCERNS:

1. Title should be “that are reminiscent of human myopathy”

2. Abstract: remove “has”: fluvastatin-treated flies has restored ClC-a expression

3. Introduction: lipoprotien should be lipoprotein; change “very crucial” to “crucial”

4. Methods: change “flies stock was purchased” to “fly stocks were purchased” “F1 progeny was used” to “F1 progeny were used”

5. Table 2: “sucrose alone or three hours” should be “sucrose alone for three hours

6. Line 210: vials not vails; line 226: 1 ml water, not Iml water

7. Microscopy methods sections: what is O.N? overnight? Please spell out. 

8. Line 310, delete the first word “Selected”

9. Line 378: “Singling detection” should be “Signal detection”

10. Length is misspelled in panel 3F and 6E. 

11. Line 548. Delete “Second”

12. Line 572. Replace “rear” with “rare”

13. Line 624. Replace “sex” with “six”

14. ClC is sometimes followed by “alpha” and sometimes by “a”. Needs to be correct and consistent. The ClC channel should be listed as a skeletal muscle channel, not a skeletal channel.

15. Figure 7A should be activity counts/30 minutes

16. Line 847 “normalized chloride expression” should be “normalized chloride channel expression”

17. Line 921-5: Duplication of text: Due to the lack of specific 921 antibodies in the market, we could not determine whether fluvastatin activates Pkcdelta 922 which could have been performed with western blot for its active and inactive form with 923 the appropriate antibodies. Furthermore, we did not show whether fluvastatin activates 924 Pkcdelta, which could be done by western blot for its active and inactive form, due to the 925 lack of specific antibodies in the market.

Author Response

(The authors gave the same response as above.)

Reviewer 3 Report

The study by Al-Sabri and collaborators is very interesting. They report an impressive amount of data regarding the skeletal muscle toxicity of fluvastatin in Drosophila. The study is well performed and the conclusions are straightforward. The results merely recapitulate what has been obtained in other species, including zebrafish, rodents, and humans. This offer a strong support to several hypotheses, including a possible role of PKC-mediated ClC-1 chloride channel inhibition and an impaired insulin signaling.

Specific comments:

From figure 1, it appears that 0.5 and 1 mM fluvastatin treatment exerted a similar effect on fly locomotion. However, 0.5 mM fluvastatin had no significant effect on ClC-a expression: please discuss.

FFigure 2 shows very important toxic effects on muscle structure after a short 5-days treatment with fluvastatin, suggesting that the dose (1 mM) may be very high. Accordingly, 0.5 mM fluvastatin has similar effects to 1 mM on fly locomotion. Is there any possibility to compare the dose used here to the human clinical doses? The effects of fluvastatin on muscle are likely concentration-dependent, and it would be of interest to investigate the dose dependence of the various effects to discern the ones more relevant for muscle toxicity. Please comment       

English needs moderate revision.

Author Response

(The authors gave the same response as above.)

Round 2

Reviewer 1 Report

The Authors have addressed the suggestions I made in the first review, and the revised manuscript is appropriatelyt revised.

Author Response

Thank you very much for your suggestions and your feedback.

Reviewer 2 Report

The authors have dealt adequately with all minor concerns as well as major concerns 1, 5, 6, 7, 8, 9, 10, 11

However, there are major issues that remain to be resolved, mostly in regard to the electron micrographs. For clarity, I am using the same numbering system as in the original review: 

MAJOR CONCERNS: 

2. I appreciate the authors revising their data and the manuscript to focus on femur muscles. The concern remains as to whether the femur contains multiple types of muscles, which could lead to incorrect conclusions about the effects of statin treatment, i.e., the differences observed result from looking at different muscle types within the femur. 

For instance, two control samples are shown in Fig. 3A. They are at the same magnification and yet the sarcomere lengths in the top panel are about half the size of the ones in the lower panel! Further, the myofilament arrangement in the top panel is unusual. Perhaps this sample is not sectioned to yield a true longitudinal image? Again, we can see this problem when comparing the control samples in panel C. 

Likewise in Fig. 4C, the two control groups show dramatically different sarcomere lengths compared to each other. 

In contrast, Supplementary Fig. 2A and Fig. 6 appear to have captured similar muscle types. 

Overall, I remain unconvinced by the sarcomere measurements for Figs. 3 and 4, as they are not convincingly comparing the same femur muscle type. I realize this criticism will not be well received by the authors, but I am trying to prevent them from publishing unfounded conclusions. Aside from more careful discernment of muscle types and orientation, the authors might consider dropping the sarcomere measurements and instead highlighting the qualitative differences in muscle structure that are observed (distorted sarcomeres, Z-line streaming, etc.). 

3. Figure 3: As per the original review: “Why do the control samples have colored arrows indicating distorted sarcomeres, Z-line streaming, etc.?” The legend states that the arrows show these abnormalities, but the arrows are also present in the control samples. Are the authors trying to show the normal and abnormal features with the same color arrows in both samples. This remains unclear in the figure legend.    

4. The authors should state the purpose of the ice treatment prior to fixation for microscopy, i.e., as a method of ensuring relaxed muscle? I am not sure this treatment would relax all muscle types, as would a calcium chelator. 

Author Response

We thank the reviewer for the comments. Please, see our response in the attached file. 
